# Understanding and Mitigating Memorization in Generative Models via Sharpness of Probability Landscapes

Dongjae Jeon [* 1]   Dueun Kim [* 2]   Albert No [2]

## Abstract

In this paper, we introduce a geometric framework to analyze memorization in diffusion models through the sharpness of the log probability density. We mathematically justify a previously proposed score-difference-based memorization metric by demonstrating its effectiveness in quantifying sharpness. Additionally, we propose a novel memorization metric that captures sharpness at the initial stage of image generation in latent diffusion models, offering early insights into potential memorization. Leveraging this metric, we develop a mitigation strategy that optimizes the initial noise of the generation process using a sharpness-aware regularization term. The code is publicly available at `https://github.com/Dongjae0324/sharpness_memorization_diffusion`.

## 1. Introduction

Recent advancements in generative models have significantly improved data generation across various domains, including image synthesis (Rombach et al., 2022), natural language processing (Achiam et al., 2023; Touvron et al., 2023), and molecular design (Alakhdar et al., 2024). Among these, diffusion models (Ho et al., 2020; Song et al., 2021c) have emerged as powerful frameworks, achieving state-of-the-art results by iteratively refining noisy samples to approximate complex data distributions (Song et al., 2021b; Saharia et al., 2022; Rombach et al., 2022).

Despite their successes, diffusion models suffer from memorization, where they replicate training samples instead of generating novel outputs (Carlini et al., 2023; Somepalli et al., 2023b; Webster, 2023). This issue is especially con-

cerning when models are trained on sensitive data, leading to privacy risks (Orrick, 2023; Joseph Saveri, 2023). Addressing memorization is critical for ensuring the responsible deployment of generative models in real-world applications.

Previous work has sought to analyze memorization using various approaches, including probability manifold analysis via Local Intrinsic Dimensionality (LID) (Ross et al., 2024; Kamkari et al., 2024), spectral characterizations (Ventura et al., 2024; Stanczuk et al., 2024), and score-based discrepancy measures (Wen et al., 2024). Additionally, attention-based methods have been used to examine memorization at the feature level (Ren et al., 2024; Chen et al., 2024).

In this work, we propose a general sharpness-based framework for understanding memorization in diffusion models. Specifically, we observe that memorization correlates with regions of sharpness in the probability landscape, which can be quantified via the Hessian of the log probability. Large negative eigenvalues of the Hessian indicate sharp, isolated regions in the learned distribution, providing a mathematically grounded explanation of memorization. Furthermore, we show that the trace-based eigenvalue statistics can serve as a robust early-stage indicator of memorization, enabling detection at the initial sampling step of generation.

Our framework also provides a justification for score based metric by interpreting it through the lens of sharpness, reinforcing its validity as a memorization detection metric. Building on this, we propose an enhanced sharpness measure with additional Hessian components, improving sensitivity, particularly at the earliest stages of sampling.

Beyond detection, we introduce an inference-time mitigation strategy that reduces memorization by selecting initial diffusion noise from regions of lower sharpness. Our method, Sharpness-Aware Initialization for Latent Diffusion (SAIL), utilizes our sharpness metric to identify initializations that avoid trajectories leading to memorization. By simply adjusting the initial noise, SAIL steers the diffusion process toward smoother probability regions, mitigating memorization without requiring retraining. Unlike prompt modifications, which can negatively affect generation quality, SAIL reduces memorization by carefully selecting the initial noise while fully preserving the conditioning inputs.

---

[*]Equal contribution  [1]Department of Computer Science, Yonsei University, Seoul, Korea [2]Department of Artificial Intelligence, Yonsei University, Seoul, Korea. Correspondence to: Albert No <albertno@yonsei.ac.kr>.

*Proceedings of the 42nd International Conference on Machine Learning*, Vancouver, Canada. PMLR 267, 2025. Copyright 2025 by the author(s).

We validate our approach through experiments on a 2D toy dataset, MNIST, and Stable Diffusion. Our results show that Hessian eigenvalues effectively differentiate memorized from non-memorized samples, and our sharpness measure provides a reliable metric for memorization detection. Additionally, we demonstrate that SAIL mitigates memorization while preserving generation quality, offering a simple yet effective solution for reducing memorization.

In summary, our key contributions are:

- We propose a sharpness-based framework for analyzing memorization in diffusion models, examining the patterns of Hessian eigenvalues and their aggregate statistics to characterize memorized samples.

- We provide a theoretical justification for the memorization detection metric introduced by Wen et al. (2024) through sharpness analysis.

- We introduce a new sharpness measure that enables early-stage memorization detection during the diffusion process.

- We propose SAIL, a simple yet effective mitigation strategy that selects initial noise leading to smoother probability regions, reducing memorization without altering model parameters or prompts.

## 2. Related works

**Understanding and Explaining Memorization.** The memorization behavior of diffusion models (DMs) has been extensively studied (Somepalli et al., 2023b; Carlini et al., 2023; Wen et al., 2024), with prior work examining contributing factors such as prompt conditioning (Somepalli et al., 2023b), data duplication (Carlini et al., 2023; Somepalli et al., 2023a), and dataset size or complexity (Gu et al., 2023). Some studies have approached this issue from a geometric standpoint, drawing on the manifold learning conjecture (Fefferman et al., 2016; Pope et al., 2021), where exact memorization is associated with data points lying on a zero-dimensional manifold (Ross et al., 2024; Ventura et al., 2024; Pidstrigach, 2022).

This geometric perspective has led to efforts to estimate Local Intrinsic Dimensionality (LID) at the sample level (Stanczuk et al., 2024; Kamkari et al., 2024; Horvat & Pfister, 2024; Wenliang & Moran, 2023; Tempczyk et al., 2022), which has been used to characterize memorization (Ross et al., 2024; Ventura et al., 2024).

While our work is inspired by prior studies, it introduces several key distinctions. Unlike approaches that define memorization in terms of overall model behavior (Yoon et al., 2023; Gu et al., 2023), we focus on sample-specific behavior manifested in the learned probability density. Although our perspective is conceptually aligned with recent geometric interpretations (Ross et al., 2024; Bhattacharjee et al., 2023), our methodology diverges fundamentally by analyzing sharpness in the learned density, without relying on assumptions about an inaccessible ground-truth distribution. In contrast to manifold-based analyses that track variations in individual feature components (Ventura et al., 2024; Achilli et al., 2024), we show that sharpness, treated as an aggregated statistic, can be effectively estimated and used for detecting memorization. Moreover, unlike LID-based methods (Ross et al., 2024) that are restricted to the final denoising step, our approach reveals that memorized samples persistently occupy high-sharpness regions throughout the diffusion process. This allows for earlier detection and targeted intervention, enabling a more proactive and interpretable strategy for mitigating memorization.

**Detecting and Mitigating Memorization.** Detecting and mitigating memorization during the generative process remains a challenging problem. Previous studies have explored various approaches to identify prompts that induce memorization in text-conditional DMs by comparing generated images to training data. For instance, Somepalli et al. (2023a) employed feature-based detectors like SSCD (Pizzi et al., 2022) and DINO (Caron et al., 2021), while Carlini et al. (2023) and Yoon et al. (2023) used calibrated $\ell_2$ distance in pixel space to quantify memorization. Webster (2023) developed both white-box and black-box attacks, analyzing edges and noise patterns in generated images.

While these methods provide valuable insights, their computational cost makes real-time detection impractical. To address this limitation, heuristic-based alternatives have been proposed. Wen et al. (2024) introduced a metric based on the magnitude of text-conditional score predictions, leveraging the observation that memorized prompts exhibit stronger text guidance. Similarly, Ren et al. (2024) identified memorization via anomalously high attention scores on specific tokens, while Chen et al. (2024) focused on patterns in end tokens of text embeddings.

Since memorization in DMs is often linked to specific text prompts, most mitigation strategies have focused on modifying prompts or adjusting attention mechanisms to reduce their influence (Wen et al., 2024; Ren et al., 2024; Ross et al., 2024). For example, Ross et al. (2024) rephrased prompts using GPT-4 to mitigate memorization. However, these interventions often degrade image quality or compromise user intent by altering model-internal components.

In contrast, our approach offers a principled and model-agnostic alternative by optimizing the initial noise input instead of modifying the text prompt or trained model parameters. By selecting initial noise that leads to smoother probability regions, our method mitigates memorization

while preserving both user prompts and model fidelity, ensuring minimal impact on generation quality.

## 3. Preliminaries

**Score-based Diffusion Models.** Diffusion models (DMs) (Sohl-Dickstein et al., 2015; Ho et al., 2020; Song et al., 2021c) generate images by iteratively refining random noise into samples that approximate the data distribution $p_0(\mathbf{x}_0)$. The process begins with the forward process, where the training data is progressively corrupted by the addition of Gaussian noise. At each timestep $t$, the conditional distribution of the noisy data is given by:

$$q_{t|0}(\mathbf{x}_t|\mathbf{x}_0) = \mathcal{N}(\mathbf{x}_t|\sqrt{\alpha_t}\mathbf{x}_0, (1-\alpha_t)\mathbf{I}),$$

where $\mathbf{x}_t$ represents the noisy data at timestep $t$, and $\alpha_t$ decreases monotonically over time in the variance-preserving case, with $\alpha_T$ becoming sufficiently small such that the resulting distribution closely resembles pure Gaussian noise:

$$q_{T|0}(\mathbf{x}_T|\mathbf{x}_0) \approx \mathcal{N}(\mathbf{0}, \mathbf{I}).$$

This process can be equivalently represented as a stochastic differential equation (SDE):

$$d\mathbf{x}_t = f(\mathbf{x}_t, t)dt + g(t)d\mathbf{w}_t,$$

where $\mathbf{w}_t$ is a standard Brownian motion.

The reverse process, which reconstructs the data distribution $p_0(\mathbf{x}_0)$ from noise, is then formulated as:

$$d\mathbf{x}_t = \left[ f(\mathbf{x}_t, t) - g^2(t)\nabla_{\mathbf{x}_t}\log p_t(\mathbf{x}_t) \right]dt + g(t)d\bar{\mathbf{w}}_t,$$

where $\bar{\mathbf{w}}_t$ denotes a standard Brownian motion in reverse time, and $p_t(\mathbf{x}_t)$ is the marginal distribution at timestep $t$.

The only unknown term in the reverse process is the score function over timesteps, $\nabla_{\mathbf{x}_t}\log p_t(\mathbf{x}_t) := s(\mathbf{x}_t)$, which is often parameterized by a neural network with $s_\theta(\mathbf{x}_t)$.

In many applications the data $\mathbf{x}_0$ is often represented with an associated label $c$ (e.g., prompts or class labels). In these scenarios, the additional condition $c$ is incorporated into the model as $s_\theta(\mathbf{x}_t, c)$, allowing it to estimate the score of the conditional density $\nabla_{\mathbf{x}_t}\log p_t(\mathbf{x}_t|c) := s(\mathbf{x}_t, c)$ via classifier free guidance (Ho & Salimans, 2021).

**Sharpness and Hessian.** For a given function $f$ at a point $x$, the Hessian $\nabla_x^2 f(x)$ represents the matrix of second-order derivatives, encapsulating the local curvature of $f$ around $x$. The eigenvectors of the Hessian define the principal axes of this curvature, while the corresponding eigenvalues characterize the curvature along these directions. Positive eigenvalues indicate local convexity, negative eigenvalues indicate local concavity, and zero eigenvalues indicate

flatness in those directions. The magnitude of an eigenvalue reflects the steepness of the curvature, with larger absolute values indicating steeper changes in $f$.

In this work, we examine the memorization by analyzing the Hessian of $\log p_t(\mathbf{x}_t)$, which corresponds to the Jacobian of the score function. We denote it as $H(\mathbf{x}_t) := \nabla_{\mathbf{x}_t}^2 \log p_t(\mathbf{x}_t)$ for the unconditional case and $H(\mathbf{x}_t, c) := \nabla_{\mathbf{x}_t}^2 \log p_t(\mathbf{x}_t|c)$ for the conditional case. The Hessian estimated by the model is denoted as $H_\theta(\mathbf{x}_t)$ and $H_\theta(\mathbf{x}_t, c)$.

## 4. Understanding Memorization via Sharpness

### 4.1. Memorization: Sharpness in Probability Landscape

Sharpness quantifies the concentration of learned log density $\log p(\mathbf{x})$ around point $\mathbf{x}$, which can be analyzed through the eigenvalues of its Hessian matrix. Large negative eigenvalues indicate sharp peaks in the distribution, suggesting memorization of specific data points. Conversely, small magnitude or positive eigenvalues characterize broader, smoother regions that facilitate better generalization.

Local Intrinsic Dimensionality (LID) (Kamkari et al., 2024) quantifies the effective dimensionality of a point in its local neighborhood, characterizing local sample space geometry. At the final generation step ($t \approx 0$), LID serves as a memorization indicator (Ross et al., 2024). Exact Memorization (EM) shows near-zero LID, indicating pure reproduction of training samples, while Partial Memorization (PM) exhibits small but nonzero LID, reflecting limited stylistic variations. In contrast, properly generalized samples demonstrate moderate LID values, indicating more diverse representations.

While both sharpness and LID characterize curvature properties of probability density, LID is limited to analyzing sample space at $t \approx 0$, where the generated image emerges. In contrast, we extend memorization detection across all timesteps by leveraging sharpness via Hessian eigenvalues as a more versatile metric, enabling continuous monitoring throughout the generative process rather than relying solely on final output characteristics.

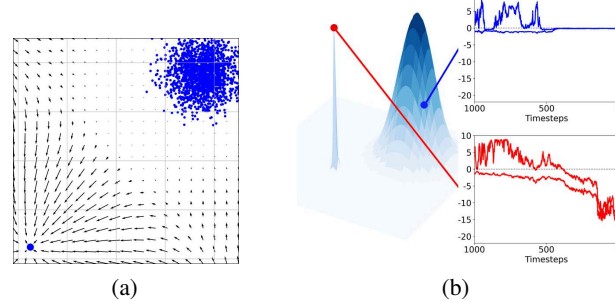

(a)      (b)

Figure 1: **(a)** Learned score vectors at final sampling step ($t = 1$), with training data points marked in blue. **(b)** Evolution of eigenvalues throughout the sampling process for a memorized (red) and non-memorized (blue) sample.

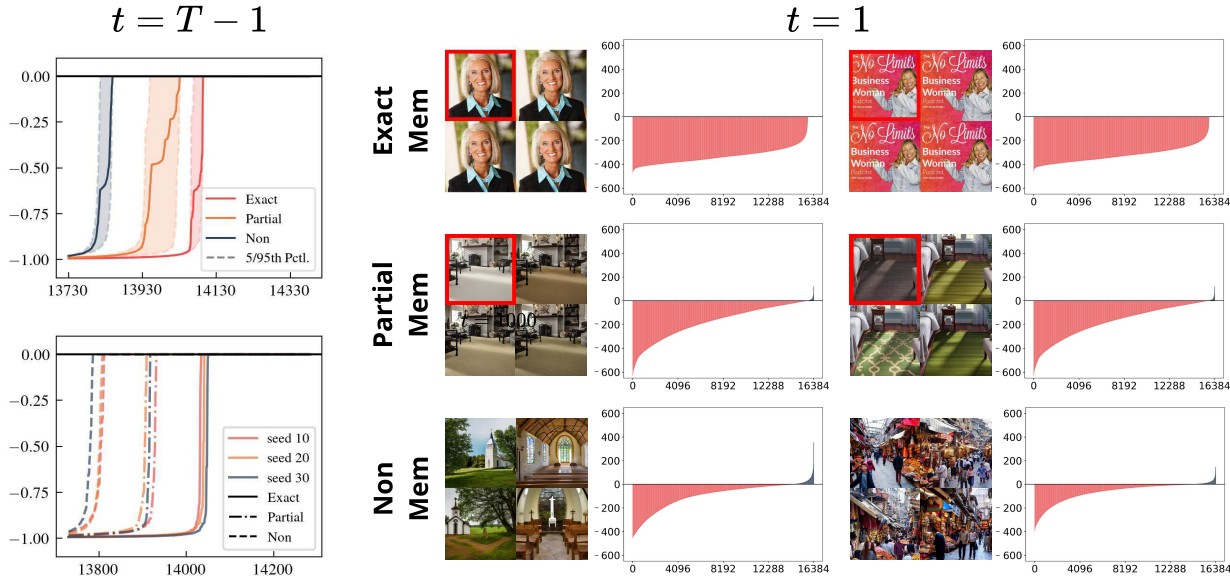

Figure 3: **Left:** Eigenvalue distribution of $H_\theta(\mathbf{x}_t, c)$ across memorization categories in Stable Diffusion v1.4 at initial sampling step ($t = T - 1$) with range clipped. **(top)** 30 prompts per category with identical initialization. **(bottom)** Fixed prompt set with three different initializations. Both plots reveal stronger memorization correlates with fewer non-negative eigenvalues. **Right:** Eigenvalue distribution of $H_\theta(\mathbf{x}_t, c)$ across memorization categories in Stable Diffusion v1.4 at final sampling step ($t = 1$). Generated images shown with original training counterparts (outlined in red). Eigenvalues are approximated via Arnoldi iteration (Arnoldi, 1951), details in Appendix A.2.

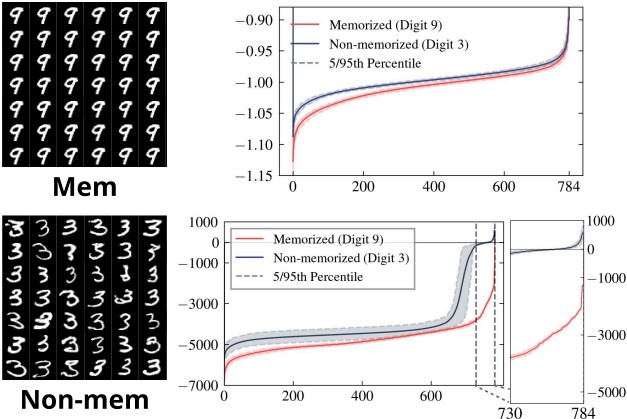

Figure 2: **Left:** Generated images for memorized (digit "9") and non-memorized (digit "3") samples. **Right:** Eigenvalue distributions for memorized (red) and non-memorized (blue) samples at initial **(top)** and final **(bottom)** sampling steps, revealing more and larger negative eigenvalues in memorized cases. Experimental details in Appendix C.

Figure 1(b) demonstrates our approach using a mixture of 2D Gaussians, where sharp peaks represent memorized distributions. From the mid stage of the denoising process, the memorized sample (red) exhibits large negative eigenvalues, indicating highly localized distributions, while the generic sample (blue) maintains near-zero eigenvalues, characterizing broader, smoother regions. Importantly, the memorized sample exhibits sharp characteristics even at intermediate timesteps, making early-stage detection possible.

To validate our approach on real data, we conduct experiments on MNIST by inducing memorization through repeated exposure to a single "9" image while maintaining all "3" images as a general class (Figure 2). The eigenvalue distributions at $t = 1$ clearly differentiate memorized from non-memorized samples: memorized samples show consistently large negative eigenvalues indicating sharp peaks, while non-memorized samples exhibit positive eigenvalues, reflecting locally convex regions that allow sample variations. Notably, these clear distributional differences emerged even at the initial sampling step ($t = T - 1$), confirming that sharpness-based memorization detection is effective from the very beginning of the generation process.

We further validate our approach on Stable Diffusion (Rombach et al., 2022), analyzing its $16,384$-dimensional latent space. Figure 3 reveals distinct patterns in both the number of non-negative eigenvalues and the magnitude of negative eigenvalues across different memorization categories (EM, PM, and non-memorized) at both initial and final sampling step. These patterns not only align with LID-based analysis at $t \approx 0$ but also demonstrate sharpness as a more generalizable memorization measure, capturing distinctive characteristics at generation onset.

### 4.2. Score Norm as a Sharpness Measure

While sharpness serves as a fundamental measure of memorization in generative models, directly computing the full spectrum of Hessian eigenvalues in high-dimensional distri-

butions, such as those in Stable Diffusion, is computationally intractable. A practical alternative is to approximate sharpness using the trace of the Hessian, a single scalar quantity that represents the sum of all eigenvalues, where large negative traces indicate sharp, highly localized regions.

A key observation is that the norm of the score function $\|s(\mathbf{x})\|$ inherently encodes information about the probability landscape's curvature. In Gaussian distributions, the score norm is directly connected to the Hessian trace, as shown in the following result. (Appendix B.2).

**Lemma 4.1.** *For a Gaussian vector* $\mathbf{x} \sim \mathcal{N}(\boldsymbol{\mu}, \boldsymbol{\Sigma})$,

$$\mathbb{E}\left[\|s(\mathbf{x})\|^2\right] = -\mathrm{tr}(H(\mathbf{x})),$$

*where* $H(\mathbf{x}) \equiv -\boldsymbol{\Sigma}^{-1}$ *is the Hessian of the log density.*

This result extends to non-Gaussian distributions under mild regularity assumptions (Appendix B.2). For theoretical clarity and ease of analysis, however, we focus on the Gaussian case. While the distribution $\mathbf{x}_t$ in diffusion processes is not strictly Gaussian at every timestep, recent studies show that at moderate to high noise levels, corresponding to the early and middle stages of the reverse process—the learned score is predominantly governed by its Gaussian component (Wang & Vastola, 2024). This approximation is further justified in latent diffusion models, where the latent variable $\mathbf{z}_t$ is explicitly regularized toward a Gaussian prior (Kingma, 2013; Rombach et al., 2022), despite the complexity of the original data distribution.

Under this Gaussian assumption at relevant sampling steps, the score norm $\|s_\theta(\mathbf{x}_t)\|^2$ provides an unbiased estimate of the negative Hessian trace $-\mathrm{tr}(H_\theta(\mathbf{x}_t))$, offering an efficient measure of the sharpness of the probability landscape.

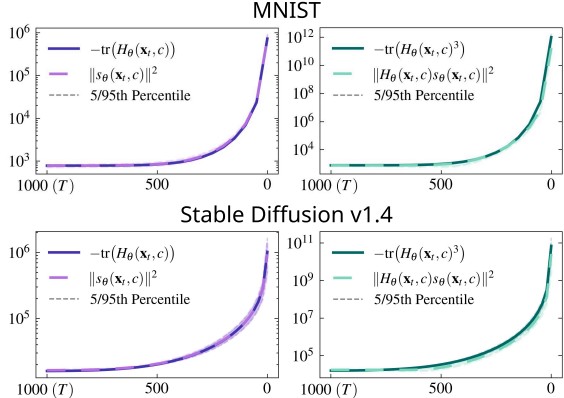

Figure 4: Empirical alignment in MNIST and Stable Diffusion between: **(left)** $-\mathrm{tr}\big(H_\theta(\mathbf{x}_t, c)\big)$ and $\|s_\theta(\mathbf{x}_t, c)\|^2$, and **(right)** $-\mathrm{tr}\big(H_\theta(\mathbf{x}_t, c)^3\big)$ and $\|H_\theta(\mathbf{x}_t, c)s_\theta(\mathbf{x}_t, c)\|^2$.

Figure 4 empirically confirms that this approximation holds reliably across datasets, including MNIST and Stable Diffusion's latent space. Surprisingly, this relationship persists

even in the later stages of the diffusion process, suggesting that score norm can serve as a computationally efficient sharpness measure throughout generation. This perspective provides a theoretical foundation for interpreting sharpness in generative models through score norm based statistic, enabling efficient memorization detection and analysis without requiring costly Hessian eigenvalue decompositions.

### 4.3. Wen's Metric as a Sharpness Measure

Wen et al. (2024) characterized memorization through the norm of difference between conditional and unconditional score functions:

$$\|s_\theta^\Delta(\mathbf{x}_t)\| := \|s_\theta(\mathbf{x}_t, c) - s_\theta(\mathbf{x}_t)\|.$$

This difference vector $s_\theta^\Delta(\mathbf{x}_t)$ determines the sampling direction in classifier-free guidance. Their approach is based on the observation that memorized prompts consistently guide generation toward specific images, resulting in larger magnitudes of $s_\theta^\Delta(\mathbf{x}_t)$ due to stronger text-driven guidance. While the theoretical foundations of this heuristic remain to be fully understood, it has proven to be one of the most effective detection metrics thus far.

Notably, the structure of $\|s_\theta^\Delta(\mathbf{x}_t)\|$ bears a strong resemblance to the score norm, which we previously identified as a measure of sharpness. This similarity hints at the possibility of interpreting Wen's metric as a sharpness measure, encapsulating the impact of conditioning on the probability distribution's curvature. To rigorously establish this connection, we proceed to analyze the Hessian of the log-density, following the same approach as in the preceding analysis.

**Lemma 4.2.** *For* $\mathbf{x} \sim \mathcal{N}(\boldsymbol{\mu}, \boldsymbol{\Sigma})$ *and* $\mathbf{x}|c \sim \mathcal{N}(\boldsymbol{\mu}_c, \boldsymbol{\Sigma}_c)$:

$$\mathbb{E}_{\mathbf{x} \sim p(\mathbf{x}|c)}\left[\|s(\mathbf{x}, c) - s(\mathbf{x})\|^2\right]$$
$$= \|H(\mathbf{x})(\boldsymbol{\mu} - \boldsymbol{\mu}_c)\|^2 + \mathrm{tr}((H(\mathbf{x}) - H_c(\mathbf{x}))^2 H_c^{-1}(\mathbf{x})),$$

*where* $H(\mathbf{x}) \equiv -\boldsymbol{\Sigma}^{-1}$ *and* $H_c(\mathbf{x}) \equiv -\boldsymbol{\Sigma}_c^{-1}$.

*Additionally, when* $\boldsymbol{\Sigma}$ *and* $\boldsymbol{\Sigma}_c$ *commute (i.e.,* $\boldsymbol{\Sigma}\boldsymbol{\Sigma}_c = \boldsymbol{\Sigma}_c\boldsymbol{\Sigma}$*) and mean vectors are the same (* $\boldsymbol{\mu} = \boldsymbol{\mu}_c$*), this reduces to*

$$\mathbb{E}_{\mathbf{x} \sim p(\mathbf{x}|c)}\left[\|s(\mathbf{x}, c) - s(\mathbf{x})\|^2\right] = \sum_{i=1}^d \frac{(\lambda_i - \lambda_{i,c})^2}{\lambda_{i,c}},$$

*where* $\lambda_i, \lambda_{i,c}$ *are eigenvalues of* $H(\mathbf{x})$ *and* $H_c(\mathbf{x})$.

This result demonstrates that Wen's metric measures sharpness differences through squared eigenvalue differences of the conditional and unconditional Hessian. During early timesteps, when the latent distribution remains close to an isotropic Gaussian, this metric directly captures the extent to which conditioning induces sharpness. At later timesteps, when $\boldsymbol{\Sigma}_t$ and $\boldsymbol{\Sigma}_{t,c}$ do not generally commute, the metric

can be interpreted through generalized eigenvalues, revealing how conditioning sharpens the learned distribution in similar manner. The details are provided in Appendix A.3.

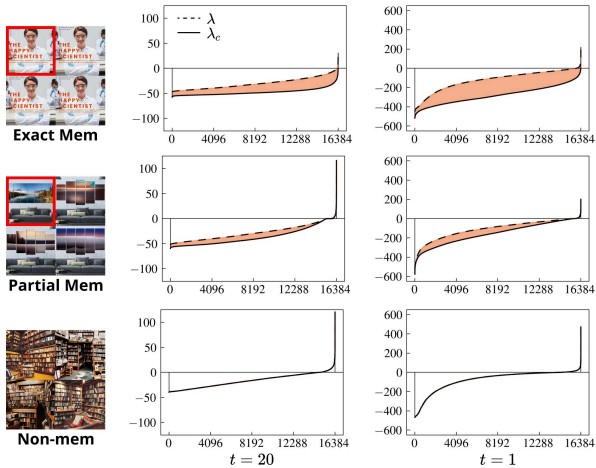

Figure 5: Eigenvalue differences between the conditional and unconditional Hessians. Memorized samples exhibit a significantly larger gap, while non-memorized samples show near-zero differences throughout. At intermediate timesteps ($t = 20$), the gap remains small but detectable, and at the final stage ($t = 1$), it widens further.

Figure 5 shows the eigenvalue disparities between conditional and unconditional Hessians across timesteps, revealing how conditioning shapes the probability distribution's geometry. For memorized samples, the eigenvalue gap is notably large, showing that conditioning creates a more constrained probability landscape. At intermediate timesteps ($t = 20$), the differences are subtle but noticeable, indicating early conditioning effects. Near the end ($t = 1$), the eigenvalue gap widens substantially, demonstrating conditioning's growing influence on the learned density. In contrast, non-memorized samples show minimal eigenvalue variations throughout, indicating little conditioning influence. These findings support our theoretical framework and confirm Wen's metric effectively measures sharpness.

### 4.4. Upscaling Eigenvalue Statistics via Hessian

While Wen's metric reveals eigenvalue disparities at intermediate timesteps, identifying and mitigating memorization during the initial generation stage remains challenging. The probability landscape maintains a nearly uniform character since the latent distribution approximates an isotropic Gaussian, making structural sharpness differences subtle. Conventional metrics struggle to capture these fine-grained distributional variations, limiting early-stage applications.

To address this limitation, we introduce a curvature-aware scaling that enhances Wen's metric through Hessian-based weighting. By multiplying the Hessian with the score function, we amplify high-curvature directions, rendering sharp

regions more distinguishable within a smooth probability landscape. This approach significantly improves the eigenvalue gap at the earliest generation stage, advancing memorization detection in the diffusion process. The following lemma shows that the Hessian-score product provides an amplified measure of the Hessian trace, thereby increasing sensitivity to distributional sharpness.

**Lemma 4.3.** *For a Gaussian vector* $\mathbf{x} \sim \mathcal{N}(\boldsymbol{\mu}, \boldsymbol{\Sigma})$,

$$\mathbb{E}\left[\|H(\mathbf{x})s(\mathbf{x})\|^2\right] = -\mathrm{tr}((H(\mathbf{x}))^3)$$

*where* $H(\mathbf{x}) \equiv -\boldsymbol{\Sigma}$ *is the Hessian of the log density.*

This relationship, empirically verified in Figure 4, demonstrates the curvature-sensitive scaling effect of the Hessian score product. Building on this principle, we propose an enhanced version of Wen's metric that improves early-stage sensitivity through second-order sharpness characterization:

$$\|H_\theta^\Delta(\mathbf{x}_t, c)s_\theta^\Delta(\mathbf{x}_t, c)\|^2,$$

where $H_\theta^\Delta(\mathbf{x}_t, c) = H_\theta(\mathbf{x}_t, c) - H_\theta(\mathbf{x}_t)$, and $s_\theta^\Delta(\mathbf{x}_t, c) = s_\theta(\mathbf{x}_t, c) - s_\theta(\mathbf{x}_t)$.

To provide intuition, assuming identical means ($\boldsymbol{\mu} = \boldsymbol{\mu}_c$) and that $\boldsymbol{\Sigma}_t$ and $\boldsymbol{\Sigma}_{t,c}$ commute, the expected value of our metric simplifies to:

$$\mathbb{E}_{\mathbf{x}_t \sim p_t(\mathbf{x}_t|c)}\left[\|H_\theta^\Delta(\mathbf{x}_t, c)s_\theta^\Delta(\mathbf{x}_t, c)\|^2\right] = \sum_{i=1}^{d} \frac{(\lambda_i - \lambda_{i,c})^4}{\lambda_{i,c}},$$

where $\lambda_i, \lambda_{i,c}$ are eigenvalues of $H(\mathbf{x}_t)$ and $H(\mathbf{x}_t, c)$.

Compared to Wen's metric in Lemma 4.2, this refinement substantially improves sensitivity by amplifying the difference in sharpness, thereby enabling more effective detection of memorization at earlier stages.

### 4.5. Detecting Memorization in Stable Diffusion

**Experimental Setup.** To evaluate our metric, we use 500 memorized prompts identified by Webster (2023) for Stable Diffusion v1.4, and 219 prompts for v2.0. As a complementary set, we include 500 non-memorized prompts sourced from COCO (Lin et al., 2014), Lexica (Lexica, 2024), Tuxemon (HuggingFace, 2024), and GPT-4 (Achiam et al., 2023). Following Wen et al. (2024), we apply the DDIM (Song et al., 2021a) sampler with 50 inference steps.

Detection performance is assessed with two standard metrics: the Area Under the Receiver Operating Characteristic Curve (AUC) and the True Positive Rate at 1% False Positive Rate (TPR@1%FPR) with higher values preferable.

For comparison, we implement six baseline methods. Among them, Carlini et al. (2023) analyzed generation density by measuring pixel-wise $\ell_2$ distances across non-overlapping image tiles, aiming to detect memorized content based on local similarity patterns. Ren et al. (2024)

| Method | Steps | $n$ | SD v1.4 | | SD v2.0 | |
|---|---|---|---|---|---|---|
| | | | AUC | TPR@1%FPR | AUC | TPR@1%FPR |
| Tiled $\ell_2$ (Carlini et al., 2023) | 50 | 4 | 0.908 | 0.088 | 0.792 | 0.114 |
| | | 16 | 0.94 | 0.232 | 0.907 | 0.114 |
| LE (Ren et al., 2024) | 1 | 1 | 0.846 | 0.116 | 0.848 | 0 |
| | | 4 | 0.839 | 0.13 | 0.853 | 0 |
| | | 16 | 0.832 | 0.124 | 0.851 | 0 |
| AE (Ren et al., 2024) | 50 | 1 | 0.606 | 0 | 0.809 | 0 |
| | | 4 | 0.628 | 0 | 0.82 | 0 |
| | | 16 | 0.598 | 0 | 0.817 | 0 |
| BE (Chen et al., 2024) | 50 | 1 | 0.986 | 0.95 | 0.983 | 0.908 |
| | | 4 | 0.997 | 0.98 | 0.99 | 0.945 |
| | | 16 | 0.997 | 0.982 | 0.99 | 0.949 |
| $\|s_\theta^\Delta(\mathbf{x}_t)\|$ (Wen et al., 2024) | 1 | 1 | 0.976 | 0.896 | 0.948 | 0.739 |
| | | 4 | 0.992 | 0.944 | 0.98 | 0.876 |
| | | 16 | 0.99 | 0.928 | 0.983 | 0.881 |
| | 5 | 1 | 0.991 | 0.932 | 0.969 | 0.885 |
| | | 4 | 0.997 | 0.978 | 0.984 | 0.917 |
| | | 16 | **0.998** | **0.982** | 0.987 | 0.931 |
| | 50 | 1 | 0.983 | 0.948 | 0.982 | 0.904 |
| | | 4 | 0.996 | **0.982** | 0.99 | **0.949** |
| | | 16 | **0.998** | 0.98 | **0.991** | 0.945 |
| $\|H_\theta^\Delta(\mathbf{x}_T)s_\theta^\Delta(\mathbf{x}_T)\|^2$ (Ours) | 1 | 1 | 0.987 | 0.908 | 0.959 | 0.74 |
| | | 4 | **0.998** | **0.982** | **0.991** | 0.895 |

Table 1: AUC and TPR@1%FPR across detection strategies and sampling steps for Stable Diffusion (SD) v1.4 and v2.0. Here, $n$ denotes the number of generations per prompt, with results averaged over $n$. "Steps" indicates the stage along the diffusion sampling path, ranging from step 1 ($t = T - 1$) to step 50 ($t = 1$).

detected memorized samples by identifying anomalous attention score patterns in text-conditioning during sampling. Chen et al. (2024) refined Wen et al. (2024)'s metric for partial memorization by incorporating end-token masks that empirically highlight locally memorized regions.

We report detection results at sampling steps 1, 5, and 50, but only include 50-step results for methods requiring full sampling or showing significant performance gains. Additional experimental details are provided in Appendix D.1.

**Results.** Table 1 demonstrates our metric's strong performance on Stable Diffusion v1.4 and v2.0 using just a single sampling step. By upscaling curvature information via $H_\theta^\Delta(\mathbf{x}_t)$, we significantly enhance Wen et al. (2024)'s metric. With merely four generations, we achieve an AUC of 0.998 and TPR@1%FPR of 0.982, matching Wen et al. (2024)'s performance using five steps and 16 generations. Similarly, in v2.0, our approach attains an AUC of 0.991 without full-step sampling, underscoring its effectiveness.

Importantly, our metric can be efficiently computed using Hessian-vector products without explicitly forming the full Hessian matrix. Leveraging automatic differentiation frameworks such as PyTorch, a single Hessian-vector product suffices for detection, incurring minimal overhead.

## 5. Sharpness Aware Memorization Mitigation

### 5.1. Sharpness Aware Initialization Sampling

**Motivation.** In Section 4, we observed that memorized samples exhibit a sharp conditional density, $p_t(\mathbf{x}_t|c)$, even at the very beginning of the generation process (i.e., at $t = T - 1$; note that sampling proceeds in reverse order, starting from $t = T$). This is substantiated by the strong detection performance of both Wen's metric and our metric at the initial sampling step, which quantifies the sharpness gap between $p_t(\mathbf{x}_t|c)$ and $p_t(\mathbf{x}_t)$.

This phenomenon, linked to the deterministic nature of ODE samplers (a one-to-one mapping between noise and image), implies that initializations from sharper densities remain in sharper regions at each intermediate timestep of the generation process, thereby increasing the likelihood of producing memorized images. In contrast, initializations from smoother regions tend to yield non-memorized images.

Thus, we argue that sampling with noise from smoother densities could effectively mitigate memorization. While manually searching for such initializations is a straightforward approach, it becomes infeasible in high-dimensional Gaussian space due to the sheer size and complexity of the search domain. Consequently, we propose to directly optimize the initial noise $\mathbf{x}_T$ as a more scalable and systematic way to address this challenge.

**Sharpness Aware Initialization.** We propose *Sharpness-Aware Initialization for Latent Diffusion* (**SAIL**), an inference-time mitigation strategy that optimizes initializations $\mathbf{x}_T$ by minimizing the sharpness gap at the starting step ($t = T - 1$). SAIL identifies initial seeds on non-memorized sampling trajectories by selecting $\mathbf{x}_T$ from smoother regions while maintaining a reasonable density under the isotropic Gaussian prior. The objective function is defined as:

$$\|H_\theta^\Delta(\mathbf{x}_T)s_\theta^\Delta(\mathbf{x}_T)\|^2 - \alpha \log p_G(\mathbf{x}_T),$$

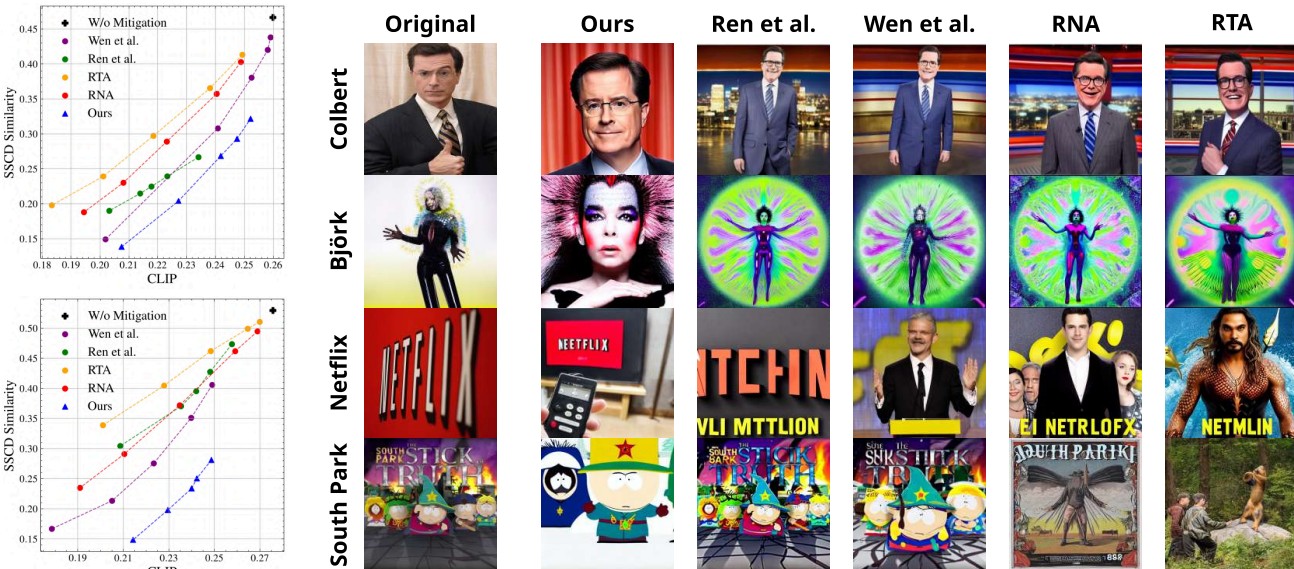

Figure 6: **Left:** Comparison of inference-time mitigation methods on SD v1.4 (top) and v2.0 (bottom), evaluated across five hyperparameter configurations per method. Lower SSCD scores indicate reduced memorization, while higher CLIP scores show better prompt-image alignment. **Right:** Qualitative comparison demonstrating SAIL's effectiveness in preserving key image details (shown adjacent to the original image), whereas baseline methods exhibit quality degradation due to modified text conditioning. Images are generated using identical random seeds, with full prompts in Appendix D.2

where $p_G$ is the density of an isotropic Gaussian distribution.

While $\|H_\theta^\Delta(\mathbf{x}_T)s_\theta^\Delta(\mathbf{x}_T)\|^2$ can be efficiently computed using Hessian-vector products, the gradient backpropagation required for optimization introduces computational overhead. To overcome the burden, we approximate the term using a Taylor expansion around $\mathbf{x}_T$:

$$\|H_\theta^\Delta(\mathbf{x}_T)s_\theta^\Delta(\mathbf{x}_T)\|^2 \approx \frac{\left\|s_\theta^\Delta\big(\mathbf{x}_T + \delta s_\theta^\Delta(\mathbf{x}_T)\big) - s_\theta^\Delta(\mathbf{x}_T)\right\|^2}{\delta^2}.$$

This leads to the final objective for SAIL:

$$\mathcal{L}_{\text{SAIL}}(\mathbf{x}_T) := \|s_\theta^\Delta\big(\mathbf{x}_T + \delta s_\theta^\Delta(\mathbf{x}_T)\big) - s_\theta^\Delta(\mathbf{x}_T)\|^2 + \alpha\|\mathbf{x}_T\|^2,$$

where $\alpha$ balances the sharpness of the density and the original likelihood. To ensure initializations remain close to the Gaussian distribution, we employ early stopping based on a threshold $\ell_{\text{thres}}$, limiting number of optimization steps.

### 5.2. Mitigating Memorization in Stable Diffusion.

**Experimental Setup.** To evaluate mitigation strategies, we use the same memorized prompt set employed in the detection experiments described in Section 4.5. However, since verifying mitigation effects requires access to training images, we exclude prompts whose corresponding training samples are unavailable. Further details are in Appendix D.

We employ two key metrics following (Wen et al., 2024; Somepalli et al., 2023a): the SSCD similarity score (Pizzi et al., 2022), which quantifies memorization by comparing model-based features of generated images to their corresponding training data, and the CLIP score (Radford et al., 2021), which evaluates prompt-image alignment. Results are averaged over five generations per prompt.

For comparison, we implement four recent mitigation algorithms. Somepalli et al. (2023b) propose Random Token Addition (RTA) and Random Number Addition (RNA), which perturb original prompts to mitigate memorization. Wen et al. (2024) introduce a method that optimizes text embeddings to reduce the influence of memorization-inducing tokens. Ren et al. (2024) propose a strategy that adjusts attention scores of text embeddings for mitigation.

For a fair comparison, all methods are evaluated using five distinct hyperparameter settings and optimized with the Adam optimizer at a learning rate of 0.05. For a detailed experimental settings, refer to Appendix D.2.

**Results.** Figure 6 (left) demonstrates that SAIL significantly improves both SSCD and CLIP metrics for Stable Diffusion v1.4 and v2.0. By optimizing the noise initialization $\mathbf{x}_T$ without altering model components like text embeddings or attention weights, SAIL effectively mitigates memorized content while preserving model behavior and user prompts, ensuring high-quality, non-memorized outputs.

The advantage of SAIL is evident in Figure 6 (right), where

it generates images that faithfully preserve key prompt details, such as celebrity names and primary objects. In contrast, methods that modify text-conditional components often reduce the influence of those components during mitigation, leading to degraded alignment with the original prompt and potentially diminishing user utility. Additional qualitative results for algorithms are provided in Appendix E.

## 6. Conclusion

We propose a sharpness-based framework for detecting and mitigating memorization in diffusion models. Our analysis identifies Hessian-based sharpness as a reliable indicator of memorization and introduces an efficient proxy based on the score norm. This perspective also provides a theoretical interpretation of the memorization detection metric proposed by Wen et al. (2024). Building on this foundation, we introduce Sharpness-Aware Initialization for Latent Diffusion (SAIL), an inference-time method that reduces memorization by selecting low-sharpness initial noise. Experiments on synthetic 2D data, MNIST, and Stable Diffusion demonstrate that our approach enables early detection and effective mitigation, all without degrading generation quality.

## Acknowledgement

This work was supported in part by Institute of Information & communications Technology Planning & Evaluation (IITP) grant funded by the Korea government(MSIT) (No. RS-2024-00457882, AI Research Hub Project), the Ministry of Science and ICT (MSIT), South Korea, under the Information Technology Research Center (ITRC) Support Program (IITP-2025-RS-2022-00156295), and IITP grant funded by the Korean Government (MSIT) (No. RS-2020-II201361, Artificial Intelligence Graduate School Program (Yonsei University)).

## Impact Statement

Our work aims to advance the understanding and mitigation of memorization in diffusion models, a phenomenon closely tied to potential privacy risks. By proposing a framework to detect and reduce memorization, we seek to enhance the responsible deployment of generative models, especially when they are trained on sensitive data. This approach could contribute positively by lowering the risk of unintentionally revealing private information.

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

# A. Additional Mathematical Details

## A.1. Second-Order Score Function

Since the Hessian of interest is simply the Jacobian of the score function, it can be directly computed using automatic differentiation from a trained diffusion model (DM). While a well-trained DM that accurately estimates scores should theoretically yield an accurate Hessian via automatic differentiation, this is not always the case in practice. Therefore, to achieve a more accurate estimation of the Hessian, the model should be parameterized and incorporate a second-order score matching loss that estimates $\nabla_{\mathbf{x}_t}^2 \log p_t(\mathbf{x}_t) \approx \nabla_{\mathbf{x}_t} s_\theta(\mathbf{x}_t) := H_\theta(\mathbf{x}_t)$ as demonstrated by Meng et al. (2021). This can be interpreted as implicit correction of the parametrized score function. To enhance numerical stability in the loss function, we adopt the loss proposed by Lu et al. (2022), an improved version of the loss utilized by Meng et al. (2021). For a fixed $t$ and given trained score function, this loss is defined as:

$$\theta^* = \arg\min_\theta \mathbb{E}_{\mathbf{x}_0, \boldsymbol{\epsilon}} \left[ \frac{1}{\sigma_t^4} \left\| \sigma_t^2 H_\theta(\mathbf{x}_t) + \mathbf{I} - \ell_1 \ell_1^\top \right\|_F^2 \right],$$

where $\ell_1(\boldsymbol{\epsilon}, \mathbf{x}_0) := \sigma_t s_\theta(\mathbf{x}_t) + \boldsymbol{\epsilon}$, $\mathbf{x}_t = \alpha_t \mathbf{x}_0 + \sigma_t \boldsymbol{\epsilon}$, $\boldsymbol{\epsilon} \sim \mathcal{N}(\mathbf{0}, \mathbf{I})$. The proposed objective is

$$\mathcal{L}_{DSM}^{(2)}(\theta) := \mathbb{E}_{t, \mathbf{x}_0, \boldsymbol{\epsilon}} \left[ \left\| \sigma_t^2 H_\theta(\mathbf{x}_t) + \mathbf{I} - \ell_1 \ell_1^\top \right\|_F^2 \right].$$

To obtain a more accurate Hessian estimate in the Toy experiment, we used $\mathcal{L} = \mathcal{L}_{DSM}(\theta) + 0.5 \mathcal{L}_{DSM}^{(2)}(\theta)$, which was simultaneously optimized using a weighted sum format. For Stable Diffusion, no additional training was performed because the original training data were not publicly available, making it difficult to retrain or fine-tune. Nevertheless, as noted in the main text, we still obtained sufficiently good results with the existing pretrained model.

## A.2. Numerical Eigenvalue Algorithm

For high-resolution image data with very large dimensions, such as in Stable Diffusion, calculating the exact Hessian and finding its eigenvalues are computationally complex and mememory inefficient. As an alternative, we employ Arnoldi iteration (Arnoldi, 1951), a numerical algorithm that leverages the efficient computation of Hessian-vector products via `torch.autograd.functional.jvp` to approximate some leading eigenvalues without forming the Hessian explicitly. In more detail, we can compute the action of the Hessian on a vector $\mathbf{v}$ efficiently using automatic differentiation. Arnoldi iteration is an algorithm derived from the Krylov subspace method that constructs an orthonormal basis $\mathbf{Q}_m = [\mathbf{q}_1, \mathbf{q}_2, \ldots, \mathbf{q}_m]$ of the Krylov subspace $K_m$, and an upper Hessenberg matrix $\mathbf{H}_m$, such that the following relationship holds:

$$\mathbf{A}\mathbf{Q}_m = \mathbf{Q}_m \mathbf{H}_m + h_{m+1,m} \mathbf{q}_{m+1} \mathbf{e}_m^\top,$$

where $\mathbf{e}_m$ is the $m$-th canonical basis vector. Since we can compute $\mathbf{A}\mathbf{q}_k$ without forming $\mathbf{A}$ explicitly, using the function jvp_func($\mathbf{q}_k$), the Arnoldi iteration proceeds as follows. First, we normalize the starting vector $\mathbf{b}$ to obtain $\mathbf{q}_1 = \frac{\mathbf{b}}{\|\mathbf{b}\|_2}$. Then, for each iteration $k = 1$ to $m$, we compute:

$$\mathbf{v} = \text{jvp\_func}(\mathbf{q}_k),$$

which represents the action of $\mathbf{A}$ on $\mathbf{q}_k$. We then orthogonalize $\mathbf{v}$ against the previous basis vectors $\mathbf{q}_1, \ldots, \mathbf{q}_k$, updating $\mathbf{h}$ and $\mathbf{v}$:

$$h_{j,k} = \mathbf{q}_j^\top \mathbf{v}, \quad \mathbf{v} = \mathbf{v} - h_{j,k} \mathbf{q}_j, \quad \text{for } j = 1, \ldots, k.$$

After orthogonalization, we compute $h_{k+1,k} = \|\mathbf{v}\|_2$. If $h_{k+1,k}$ is greater than a small threshold $\varepsilon$, we normalize $\mathbf{v}$ to obtain the next basis vector $\mathbf{q}_{k+1} = \frac{\mathbf{v}}{h_{k+1,k}}$. Otherwise, the iteration terminates. The eigenvalues of $\mathbf{H}_m$(Ritz values) approximate the $m$ eigenvalues of $\mathbf{A}$. For details on the computational process of Arnoldi iteration, Please refer to the algorithm pesudo code below. The Arnoldi iteration tends to find eigenvalues with larger absolute values first because components associated with these eigenvalues dominate within the Krylov subspace. If the input matrix is symmetric, Arnoldi iteration can be simplified to Lanczos iteration (Lanczos, 1950). However, since the Lanczos iteration is very sensitive to small numerical errors breaking the symmetry, we use the general version. The computational complexity of the algorithm is $O(m^2 d)$ with space complexity $O(md)$, compared to $O(d^3)$ with $O(d^2)$ of exact derivation and eigendecomposition of Hessian. We calculate all eigenvalues for several samples for clear justification. But with just a few($m \ll d$) iterations, the difference between memorized samples and non-memorized samples reveals enough.

---

**Algorithm 1** Arnoldi Iteration using Jacobian-Vector Products

---

**Require:** Starting vector $\mathbf{b} \in \mathbb{R}^d$, number of iterations $m \leq d$,
    function jvp_func($\mathbf{v}$) that computes $\mathbf{A}\mathbf{v}$, threshold $\varepsilon$
**Ensure:** Orthonormal basis $\mathbf{Q}_m = [\mathbf{q}_1, \ldots, \mathbf{q}_m]$,
    upper Hessenberg matrix $\mathbf{H}_m \in \mathbb{R}^{m \times m}$
 1: Initialize $\mathbf{Q} \in \mathbb{R}^{d \times (m+1)}$, $\mathbf{h} \in \mathbb{R}^{(m+1) \times m}$
 2: Normalize the starting vector: $\mathbf{q}_1 = \frac{\mathbf{b}}{\|\mathbf{b}\|_2}$
 3: **for** $k = 1$ to $m$ **do**
 4:    Compute $\mathbf{v} \leftarrow$ jvp_func($\mathbf{q}_k$)
 5:    **for** $j = 1$ to $k$ **do**
 6:        Compute $h_{j,k} \leftarrow \mathbf{q}_j^\top \mathbf{v}$
 7:        Update $\mathbf{v} \leftarrow \mathbf{v} - h_{j,k}\mathbf{q}_j$
 8:    **end for**
 9:    Compute $h_{k+1,k} \leftarrow \|\mathbf{v}\|_2$
10:    **if** $h_{k+1,k} > \varepsilon$ **then**
11:        Normalize $\mathbf{q}_{k+1} \leftarrow \frac{\mathbf{v}}{h_{k+1,k}}$
12:    **else**
13:        **break** {Terminate iteration}
14:    **end if**
15: **end for**
16: Adjust $\mathbf{H}_m$ by removing the last row of $\mathbf{h}$
17: **return** $\mathbf{Q}_m = [\mathbf{q}_1, \ldots, \mathbf{q}_m]$,
    $\mathbf{H}_m = [h_{i,j}]_{i=1,\ldots,m;\, j=1,\ldots,m}$

---

### A.3. Generalized Eigenvalue Analysis of Score Difference

In the main text, we demonstrated that Wen et al. (2024)'s metric can be expressed in terms of Hessian eigenvalue differences. Here, we provide a more detailed derivation, including the non-commuting case, which requires the use of *generalized eigenvalues*.

Consider two Gaussian distributions: the unconditional distribution

$$\mathcal{N}(\boldsymbol{\mu}, \boldsymbol{\Sigma}_t),$$

and the conditional distribution

$$\mathcal{N}(\boldsymbol{\mu}_c, \boldsymbol{\Sigma}_{t,c}).$$

For simplicity, we assume the means are identical ($\boldsymbol{\mu} = \boldsymbol{\mu}_c$) and focus on the effect of covariance differences. Wen's metric approximately measures

$$\big\| s(\mathbf{x}_t, c) - s(\mathbf{x}_t) \big\|,$$

Through direct calculation, the expected squared difference in these scores is

$$\mathbb{E}_{\mathbf{x}_t \sim p(\mathbf{x}_t | c)}\Big[\big\| s(\mathbf{x}_t, c) - s(\mathbf{x}_t) \big\|^2\Big] = \mathrm{tr}\Big[\big(\boldsymbol{\Sigma}_t^{-1} - \boldsymbol{\Sigma}_{t,c}^{-1}\big)^2 \boldsymbol{\Sigma}_{t,c}\Big].$$

When $\boldsymbol{\Sigma}_t \boldsymbol{\Sigma}_{t,c} = \boldsymbol{\Sigma}_{t,c} \boldsymbol{\Sigma}_t$, this trace simplifies to a sum of squared eigenvalue differences:

$$\sum_i \frac{(\lambda_i - \lambda_{i,c})^2}{\lambda_{i,c}}.$$

However, when $\boldsymbol{\Sigma}_t$ and $\boldsymbol{\Sigma}_{t,c}$ do not commute, their respective eigen-decompositions cannot be directly aligned. In this case, we introduce *generalized eigenvalues* $\lambda$ by solving

$$\boldsymbol{\Sigma}_t^{-1}\mathbf{v} = \lambda \boldsymbol{\Sigma}_{t,c}^{-1}\mathbf{v}.$$

Intuitively, these $\lambda$ measure how $\mathbf{\Sigma}_t$ transforms relative to $\mathbf{\Sigma}_{t,c}$ along each direction. Note that we can rewrite the trace term in the expectation as

$$
\begin{aligned}
\mathrm{tr}\big[(\mathbf{\Sigma}_t^{-1} - \mathbf{\Sigma}_{t,c}^{-1})^2\, \mathbf{\Sigma}_{t,c}\big] &= \mathrm{tr}\Big[\big(\mathbf{\Sigma}_{t,c}^{-1/2}(\mathbf{\Sigma}_{t,c}^{1/2}\mathbf{\Sigma}_t^{-1}\mathbf{\Sigma}_{t,c}^{1/2} - \mathbf{I})\mathbf{\Sigma}_{t,c}^{-1/2}\big)^2 \mathbf{\Sigma}_{t,c}\Big] \\
&= \mathrm{tr}\Big[\big(\mathbf{\Sigma}_{t,c}^{1/2}\, \mathbf{\Sigma}_t^{-1}\, \mathbf{\Sigma}_{t,c}^{1/2} \,-\, \mathbf{I}\big)^2 \mathbf{\Sigma}_{t,c}^{-1}\Big] \\
&= \sum_{k=1}^{d}\sum_{j=1}^{d}(\lambda_k - 1)^2\, w_{k,j},
\end{aligned}
$$

where $w_{k,j}$ are weights induced by $\mathbf{\Sigma}_{t,c}^{-1}$. The $\lambda_k$s are eigenvalues of $\mathbf{\Sigma}_{t,c}^{1/2}\mathbf{\Sigma}_t^{-1}\mathbf{\Sigma}_{t,c}^{1/2}$. Since

$$
\mathbf{\Sigma}_{t,c}^{1/2}\mathbf{\Sigma}_t^{-1}\mathbf{\Sigma}_{t,c}^{1/2}\mathbf{y} = \lambda\mathbf{y},
$$

setting $\mathbf{v} = \mathbf{\Sigma}_{t,c}^{1/2}\mathbf{y}$ yields

$$
\mathbf{\Sigma}_{t,c}^{1/2}\mathbf{\Sigma}_t^{-1}\mathbf{v} = \lambda\mathbf{\Sigma}_{t,c}^{-1/2}\mathbf{v} \implies \mathbf{\Sigma}_t^{-1}\mathbf{v} = \lambda\mathbf{\Sigma}_{t,c}^{-1}\mathbf{v}.
$$

When $\lambda < 1$, since

$$
\frac{\mathbf{v}^\top\mathbf{\Sigma}_t^{-1}\mathbf{v}}{\mathbf{v}^\top\mathbf{\Sigma}_{t,c}^{-1}\mathbf{v}} = \lambda,
$$

the unconditional covariance $\mathbf{\Sigma}_t$ is effectively larger (less sharp) in that eigen-direction, indicating that the conditional distribution is sharper by comparison. Consequently, the difference $\| s(\mathbf{x}_t, c) - s(\mathbf{x}_t)\|$ encodes how much sharper (or flatter) the conditional distribution is along each generalized eigenvector. This extends the simpler commuting-case result discussed in the main text, providing a more general interpretation of Wen's metric in terms of non-commuting covariances.

### A.4. Score Difference Norm and Fisher-Rao Equivalence

Here, we show that for small perturbations $\delta\mathbf{\Sigma}_t$, the local geometry prescribed by the Fisher-Rao metric coincides with that implied by the expected squared norm of the score difference. Specifically, let $\mathbf{\Sigma}_{t,c} = \mathbf{\Sigma}_t + \delta\mathbf{\Sigma}_t$ with $\|\delta\mathbf{\Sigma}_t\| \ll 1$. By expanding both the Fisher-Rao distance and the expected score-difference norm in powers of $\delta\mathbf{\Sigma}_t$ up to second order, we find that their expansions match exactly in this limit. Importantly, this matching of expansions implies that the derivatives of the two measures with respect to $\mathbf{\Sigma}_t$ also coincide (i.e., as $\delta\mathbf{\Sigma}_t \to 0$). In other words, the local (infinitesimal) curvature on the covariance manifold-in other words, the Riemannian structure encoded by the second-order terms-is the same whether we measure distance via Fisher-Rao or via the expected score-difference norm. Consequently, both metrics capture how conditioning sharpens the learned distribution in precisely the same way under small perturbations, thereby confirming that the two approaches share the same local geometry on the Gaussian covariance manifold.

The Fisher-Rao (or affine-invariant) distance (A. Micchelli & Noakes, 2005) between $\mathbf{\Sigma}_t$ and $\mathbf{\Sigma}_{t,c}$ is

$$
d_{\mathrm{FR}}(\mathbf{\Sigma}_t, \mathbf{\Sigma}_{t,c})^2 \;=\; \Big\|\log\!\Big(\mathbf{\Sigma}_{t,c}^{-1/2}\, \mathbf{\Sigma}_t\, \mathbf{\Sigma}_{t,c}^{-1/2}\Big)\Big\|_F^2.
$$

In particular, we show that for small perturbations in $\mathbf{\Sigma}_t$, the expected norm of the score difference coincides with this squared Fisher-Rao distance up to second order. Define a small perturbation on $\mathbf{\Sigma}_t$ as $\delta\mathbf{\Sigma}_t$, where $\delta$ can be arbitrarily small. Let $\mathbf{\Sigma}_{t,c} = \mathbf{\Sigma}_t + \delta\mathbf{\Sigma}_t$, with $\mathbf{\Sigma}_t \succ 0$ and $\|\delta\mathbf{\Sigma}_t\| \ll 1$ so that $\mathbf{\Sigma}_{t,c}$ remains positive-definite. Define

$$
H^\Delta \;:=\; \mathbf{\Sigma}_t^{-1} \;-\; \mathbf{\Sigma}_{t,c}^{-1}.
$$

Since $s(\mathbf{x}_t, c) = -\mathbf{\Sigma}_{t,c}^{-1}(\mathbf{x}_t - \boldsymbol{\mu})$ and $s(\mathbf{x}_t) = -\mathbf{\Sigma}_t^{-1}(\mathbf{x}_t - \boldsymbol{\mu})$, their difference is

$$
s^\Delta(\mathbf{x}_t) \;=\; H^\Delta\,(\mathbf{x}_t - \boldsymbol{\mu}).
$$

Hence,

$$
\mathbb{E}_{\mathbf{x}_t \sim p_t(\mathbf{x}_t|c)}\Big[\|s^\Delta(\mathbf{x}_t)\|^2\Big] \;=\; \mathrm{tr}\big((H^\Delta)^2\, \mathbf{\Sigma}_{t,c}\big).
$$

Next, expand $\Sigma_{t,c}^{-1} = (\Sigma_t + \delta\Sigma_t)^{-1}$ using the Neumann series. Up to $O(\|\delta\Sigma_t\|^2)$,

$$\Sigma_{t,c}^{-1} \approx \Sigma_t^{-1} - \Sigma_t^{-1}\,\delta\Sigma_t\,\Sigma_t^{-1},$$

which yields

$$H^\Delta \approx \Sigma_t^{-1}\,\delta\Sigma_t\,\Sigma_t^{-1}, \quad (H^\Delta)^2 \approx \Sigma_t^{-1}\,\delta\Sigma_t\,\Sigma_t^{-1}\,\delta\Sigma_t\,\Sigma_t^{-1}.$$

Then,

$$(H^\Delta)^2\,\Sigma_{t,c} \approx \Sigma_t^{-1}\,\delta\Sigma_t\,\Sigma_t^{-1}\,\delta\Sigma_t,$$

so

$$\mathrm{tr}\big[(H^\Delta)^2\,\Sigma_{t,c}\big] \approx \mathrm{tr}\Big(\Sigma_t^{-1}\,\delta\Sigma_t\,\Sigma_t^{-1}\,\delta\Sigma_t\Big).$$

On the other hand, consider the Fisher-Rao distance:

$$d_{\mathrm{FR}}^2(\Sigma_t, \Sigma_{t,c}) \approx \big\|\log\big(\Sigma_{t,c}^{-1/2}\,\Sigma_t\,\Sigma_{t,c}^{-1/2}\big)\big\|_F^2.$$

Define $\mathbf{A} := \Sigma_{t,c}^{-1/2}\,\Sigma_t\,\Sigma_{t,c}^{-1/2}$. Since $\delta\Sigma_t$ is small, we can write $\mathbf{A} \approx \mathbf{I} + \mathbf{X}$ with $\|\mathbf{X}\| \ll 1$. Then,

$$\log(\mathbf{A}) \approx \mathbf{X}, \quad \|\log(\mathbf{A})\|_F^2 \approx \|\mathbf{X}\|_F^2.$$

It can be shown (via expansion in $\delta\Sigma_t$) that $\|\mathbf{X}\|_F^2$ matches $\mathrm{tr}(\Sigma_t^{-1}\,\delta\Sigma_t\,\Sigma_t^{-1}\,\delta\Sigma_t)$ up to second order, leading to

$$d_{\mathrm{FR}}^2(\Sigma_t, \Sigma_{t,c}) \approx \mathrm{tr}\Big(\Sigma_t^{-1}\,\delta\Sigma_t\,\Sigma_t^{-1}\,\delta\Sigma_t\Big).$$

Hence, combining the two expansions shows:

$$\mathbb{E}_{\mathbf{x}_t \sim p_t(\mathbf{x}_t|c)}\Big[\|s^\Delta(\mathbf{x}_t)\|^2\Big] \quad \text{and} \quad d_{\mathrm{FR}}^2(\Sigma_t, \Sigma_{t,c})$$

coincide to second order in $\|\delta\Sigma_t\|$. Thus, in the small-perturbation limit, the expected value of the squared norm of the score difference encodes the same information as the Fisher-Rao distance, affirming that Wen's metric indeed captures how conditioning sharpens the learned distribution from a Riemannian perspective.

## B. Proofs

### B.1. Proof of Lemma 4.1

**State.** *For a Gaussian vector* $\mathbf{x} \sim \mathcal{N}(\boldsymbol{\mu}, \Sigma)$,

$$\mathbb{E}\big[\|s(\mathbf{x})\|^2\big] = -\mathrm{tr}\big(H(\mathbf{x})\big),$$

*where* $H(\mathbf{x}) \equiv -\Sigma^{-1}$ *is the Hessian of the log-density.*

*Proof.* A Gaussian log-density has

$$\log p(\mathbf{x}) = -\tfrac{1}{2}(\mathbf{x} - \boldsymbol{\mu})^\top \Sigma^{-1}(\mathbf{x} - \boldsymbol{\mu}) + \text{const.},$$

so $H(\mathbf{x}) = -\Sigma^{-1}$ and $s(\mathbf{x}) = -\Sigma^{-1}(\mathbf{x} - \boldsymbol{\mu})$. Then

$$\|s(\mathbf{x})\|^2 = (\mathbf{x} - \boldsymbol{\mu})^\top \Sigma^{-2}(\mathbf{x} - \boldsymbol{\mu}).$$

Taking expectation, using $\mathbb{E}[(\mathbf{x} - \boldsymbol{\mu})^\top A\,(\mathbf{x} - \boldsymbol{\mu})] = \mathrm{tr}(A\,\Sigma))$, we get $\mathbb{E}[\|s(\mathbf{x})\|^2] = \mathrm{tr}(\Sigma^{-1}) = -\mathrm{tr}(H(\mathbf{x}))$. $\square$

This result generalizes to non-Gaussian distributions under weak regularity conditions (Hyvärinen, 2005). Although we chose the Gaussian assumption to facilitate theoretical extensions and applications, we will still present the original generalization here.

**B.2. Generalization of Lemma 4.1**

**State.** *For a random vector* $\mathbf{x} \sim p(\mathbf{x})$ *with regularity conditions* $\mathbb{E}\left[\|s(\mathbf{x})\|^2\right] < \infty$ *and* $\lim_{\|x\| \to \infty} p(\mathbf{x})s(\mathbf{x}) = \mathbf{0}$,

$$\mathbb{E}\left[\|s(\mathbf{x})\|^2\right] = -\mathbb{E}\left[\mathrm{tr}(H(\mathbf{x}))\right].$$

*Proof.* Write $s_i(\mathbf{x}) = \partial_{x_i} \log p(\mathbf{x})$. Because $s_i\, p = \partial_{x_i} p$,

$$\mathbb{E}\big[\|s(\mathbf{x})\|^2\big] = \sum_{i=1}^{d} \int s_i(\mathbf{x})\, \partial_{x_i} p(\mathbf{x})\, d\mathbf{x}.$$

For each $i$ integrate by parts:

$$\int s_i\, \partial_{x_i} p = \int \partial_{x_i}[p\, s_i]\, d\mathbf{x} - \int p\, \partial_{x_i} s_i\, d\mathbf{x}.$$

The first term is a surface integral over the sphere of radius $R$; by the assumed boundary condition it vanishes as $R \to \infty$. Hence $\int s_i\, \partial_{x_i} p = -\int p\, \partial_{x_i} s_i$. Summing over $i$ gives

$$\mathbb{E}\big[\|s(\mathbf{x})\|^2\big] = -\int p(\mathbf{x}) \sum_{i=1}^{d} \partial_{x_i} s_i(\mathbf{x})\, d\mathbf{x} = -\mathbb{E}\big[\mathrm{tr}(H(\mathbf{x}))\big].$$

$\square$

**B.3. Proof of Lemma 4.2**

**State.** *For* $\mathbf{x} \sim \mathcal{N}(\boldsymbol{\mu}, \boldsymbol{\Sigma})$ *and* $\mathbf{x}|c \sim \mathcal{N}(\boldsymbol{\mu}_c, \boldsymbol{\Sigma}_c)$:

$$\mathbb{E}_{\mathbf{x} \sim p(\mathbf{x}|c)}\big[\|s(\mathbf{x}, c) - s(\mathbf{x})\|^2\big] = \|H(\mathbf{x})(\boldsymbol{\mu} - \boldsymbol{\mu}_c)\|^2 + \mathrm{tr}\big[(H(\mathbf{x}) - H_c(\mathbf{x}))^2 H_c^{-1}(\mathbf{x})\big],$$

*where* $H(\mathbf{x}) \equiv -\boldsymbol{\Sigma}^{-1}$ *and* $H_c(\mathbf{x}) \equiv -\boldsymbol{\Sigma}_c^{-1}$.

*Additionally, if* $\boldsymbol{\Sigma}\boldsymbol{\Sigma}_c = \boldsymbol{\Sigma}_c\boldsymbol{\Sigma}$ *and* $\boldsymbol{\mu} = \boldsymbol{\mu}_c$, *then*

$$\mathbb{E}_{\mathbf{x} \sim p(\mathbf{x}|c)}\big[\|s(\mathbf{x}, c) - s(\mathbf{x})\|^2\big] = \sum_{i=1}^{d} \frac{(\lambda_i - \lambda_{i,c})^2}{\lambda_{i,c}},$$

*where* $\lambda_i, \lambda_{i,c}$ *are eigenvalues of* $H(\mathbf{x})$ *and* $H_c(\mathbf{x})$.

*Proof.* Let $s(\mathbf{x}) = -\boldsymbol{\Sigma}^{-1}(\mathbf{x} - \boldsymbol{\mu})$ and $s(\mathbf{x}, c) = -\boldsymbol{\Sigma}_c^{-1}(\mathbf{x} - \boldsymbol{\mu}_c)$ denote the Gaussian score functions for the unconditional and conditional distributions. Then

$$s(\mathbf{x}, c) - s(\mathbf{x}) = -\boldsymbol{\Sigma}_c^{-1}(\mathbf{x} - \boldsymbol{\mu}_c) + \boldsymbol{\Sigma}^{-1}(\mathbf{x} - \boldsymbol{\mu}).$$

Taking the expectation,

$$
\begin{aligned}
\mathbb{E}_{\mathbf{x} \sim p(\mathbf{x}|c)}\big[\| -\boldsymbol{\Sigma}_c^{-1}(\mathbf{x} - \boldsymbol{\mu}_c) + \boldsymbol{\Sigma}^{-1}(\mathbf{x} - \boldsymbol{\mu})\|^2\big] =\ & \mathbb{E}_{\mathbf{x} \sim p(\mathbf{x}|c)}\big[\|\boldsymbol{\Sigma}_c^{-1}(\mathbf{x} - \boldsymbol{\mu}_c)\|^2\big] \\
& + \mathbb{E}_{\mathbf{x} \sim p(\mathbf{x}|c)}\big[\|\boldsymbol{\Sigma}^{-1}(\mathbf{x} - \boldsymbol{\mu})\|^2\big] \\
& - \mathbb{E}_{\mathbf{x} \sim p(\mathbf{x}|c)}\big[(\mathbf{x} - \boldsymbol{\mu}_c)^\top \boldsymbol{\Sigma}_c^{-1}\boldsymbol{\Sigma}^{-1}(\mathbf{x} - \boldsymbol{\mu})\big] \\
& - \mathbb{E}_{\mathbf{x} \sim p(\mathbf{x}|c)}\big[(\mathbf{x} - \boldsymbol{\mu})^\top \boldsymbol{\Sigma}^{-1}\boldsymbol{\Sigma}_c^{-1}(\mathbf{x} - \boldsymbol{\mu}_c)\big] \\
=\ & \mathrm{tr}(\boldsymbol{\Sigma}_c^{-1}) + \mathrm{tr}(\boldsymbol{\Sigma}^{-2}\boldsymbol{\Sigma}_c) + (\boldsymbol{\mu}_c - \boldsymbol{\mu})^\top \boldsymbol{\Sigma}^{-2}(\boldsymbol{\mu}_c - \boldsymbol{\mu}) \\
& - \mathrm{tr}(\boldsymbol{\Sigma}_c^{-1}\boldsymbol{\Sigma}^{-1}\boldsymbol{\Sigma}_c) - \mathrm{tr}(\boldsymbol{\Sigma}^{-1}\boldsymbol{\Sigma}_c^{-1}\boldsymbol{\Sigma}_c) \\
=\ & \|\boldsymbol{\Sigma}^{-1}(\boldsymbol{\mu}_c - \boldsymbol{\mu})\|^2 + \mathrm{tr}\big((\boldsymbol{\Sigma}^{-1} - \boldsymbol{\Sigma}_c^{-1})^2\boldsymbol{\Sigma}_c\big).
\end{aligned}
$$

if $\boldsymbol{\mu} = \boldsymbol{\mu}_c$, and $\boldsymbol{\Sigma}\boldsymbol{\Sigma}_c = \boldsymbol{\Sigma}_c\boldsymbol{\Sigma}$ so that $\boldsymbol{\Sigma}^{-1}$ and $\boldsymbol{\Sigma}_c^{-1}$ are simultaneously diagonalizable as $\boldsymbol{\Sigma}^{-1} = \mathbf{Q}\boldsymbol{\Lambda}\mathbf{Q}^\top$ and $\boldsymbol{\Sigma}_c^{-1} = \mathbf{Q}\boldsymbol{\Lambda}_c\mathbf{Q}^\top$, the trace term becomes

$$\mathrm{tr}(\boldsymbol{\Sigma}^{-1} - \boldsymbol{\Sigma}_c^{-1})^2\boldsymbol{\Sigma}_c) = \mathrm{tr}(\mathbf{Q}(\boldsymbol{\Lambda} - \boldsymbol{\Lambda}_c)^2\boldsymbol{\Lambda}_c^{-1}\mathbf{Q}^\top) = \mathrm{tr}((\boldsymbol{\Lambda} - \boldsymbol{\Lambda}_c)^2\boldsymbol{\Lambda}_c^{-1})$$
$$= \sum_{i=1}^d \frac{(\lambda_i - \lambda_{i,c})^2}{\lambda_{i,c}}.$$

$\square$

### B.4. Proof of Lemma 4.3

**State.** *For a Gaussian vector* $\mathbf{x} \sim \mathcal{N}(\boldsymbol{\mu}, \boldsymbol{\Sigma})$,

$$\mathbb{E}\left[\|H(\mathbf{x})s(\mathbf{x})\|^2\right] = -\mathrm{tr}((H(\mathbf{x}))^3)$$

*where* $H(\mathbf{x}) \equiv -\boldsymbol{\Sigma}$ *is the Hessian of the log density.*

*Proof.* As $H(\mathbf{x}) = -\boldsymbol{\Sigma}^{-1}$ and $s(\mathbf{x}) = -\boldsymbol{\Sigma}^{-1}(\mathbf{x} - \boldsymbol{\mu})$,

$$\mathbb{E}\left[\|H(\mathbf{x})s(\mathbf{x})\|^2\right] = \mathbb{E}\left[(\mathbf{x} - \boldsymbol{\mu})^\top\boldsymbol{\Sigma}^{-4}(\mathbf{x} - \boldsymbol{\mu})\right] = \mathrm{tr}(\boldsymbol{\Sigma}^{-3}) = -\mathrm{tr}(H(\mathbf{x})^3).$$

$\square$

## C. Details of the Toy Experiments

This section provides additional details on the 2D and MNIST experiments discussed in Section 4.1. For both experiments, we use the DDPM (Ho et al., 2020) framework with the DDIM (Song et al., 2021a) sampler, employing 500 sampling steps. Additionally, to obtain a more accurate estimate of the Hessian (Jacobian of the score function), we utilize the second-order score matching loss proposed by Lu et al. (2022) during model training. Refer to Appendix A.1 for details.

**2D Mixture of Gaussian Experiment.** We use a mixture of Gaussians with two modes equidistant from zero but with differing covariance scales. One mode is designed with an extremely small covariance to induce a sharp peak, representing memorization, while the other mode has a larger covariance for the opposite case.

The mixture ratio between the two modes is 5:95, with a dataset comprising 3,000 samples in total. Empirically, we observed that only samples from the mode with extremely small covariance exhibited memorization, indicated by extremely small $\ell_2$ distances between the generated samples and training samples.

**MNIST Experiment.** In the MNIST experiment, we use two digits: "3" for the generalized case and "9" for the memorized case, with 3,000 samples each. Classifier-free guidance (Ho & Salimans, 2021) (CFG) is employed, training the unconditional score function $s(\mathbf{x}_t)$ with a probability $p = 0.2$ using all 6,000 samples.

For $s(\mathbf{x}_t, c)$, all samples of digit "3" are used to enable generalization and diversity, while a single sample of digit "9" (duplicated 100 times) is used to collapse the model's output for this digit into a single conditioned image. Sampling is performed with a guidance scale of 5. As expected, even with CFG, the model generates only a single image for digit "9," while producing diverse outputs for digit "3."

In Figure 2, for the non-memorized case, we sample 1,000 images and select the top 500 samples with the largest pairwise $\ell_2$ distances from training samples to highlight cases clearly deviating from memorization. For the memorized case, as all images collapse into a single image, we sample 500 outputs without comparing $\ell_2$ distances.

## D. Details of the Stable Diffusion Experiments

This section describes the experimental setups for the Stable Diffusion experiments presented in Section 4.5 and Section 5. We provide a detailed overview of the configurations, including the specific prompts used and the implementation details of the baseline methods.

**Models.** We use Stable Diffusion v1.4 and v2.0, the same versions in which memorized prompts were identified by (Wen et al., 2024). For both detection and mitigation experiments, we use the DDIM sampler (Song et al., 2021a) with 50 sampling steps following Wen et al. (2024); Ross et al. (2024).

**Prompt Configuration.**

- **Memorized Prompts:** Following recent studies (Wen et al., 2024; Ren et al., 2024; Ross et al., 2024; Chen et al., 2024), we use memorized prompts identified by Webster (2023) in our experiments. Webster (2023) categorized memorized prompts into three types: 1) *Matching Verbatim (MV)*: Generated images are exact pixel-by-pixel matches with the original paired training image. 2) *Template Verbatim (TV)*: Generated images partially resemble the training image but may differ in attributes like color or style. 3) *Retrieval Verbatim (RV)*: Generated images memorize certain training images but are associated with prompts different from the original captions. The categorization of MV, TV, and RV considers both the memorized portions of generated images and their associations with specific prompt-image pairs. For instance, a prompt generating a pixel-perfect match to a training image is classified as RV, not MV, if the prompt differs from the original training caption. However, in our study, these categories are used to differentiate between images that are exact pixel-level matches and those that replicate specific attributes, such as style or color. For simplicity, we refer exact matches as **Exact Memorization (EM)** and partial matches as **Partial Memorization (PM)**, without considering their caption associations.

  For detection experiments, we combine prompts from all categories, resulting in a total of 500 memorized prompts for Stable Diffusion v1.4, identical to the prompts used by Wen et al. (2024), and 219 prompts for v2.0.

  While detection experiments only require a prompt set, mitigation experiments necessitate access to the original training images to evaluate SSCD (Pizzi et al., 2022) scores. Consequently, prompts without accessible training images are excluded, resulting in 454 prompts for v1.4 and 202 prompts for v2.0.

- **Non-memorized Prompts:** To ensure a diverse distribution of non-memorized prompts, we compile a total of 500 prompts drawn from COCO (Lin et al., 2014), Lexica (Lexica, 2024), Tuxemon (HuggingFace, 2024), and GPT-4 (Achiam et al., 2023). Specifically, the GPT-4 prompts are a random subset of those used by (Ren et al., 2024).

### D.1. Memorization Detection

**Details for Baseline Methods.** We provide details of each baseline detection algorithm.

- **Tiled $\ell_2$ distance**: Building on the insight that memorized prompts produce similar generations regardless of their initializations, Carlini et al. (2023) propose examining generation density by analyzing multiple generated images for a given prompt using pairwise $\ell_2$ distances in pixel space. To address false positives from similar backgrounds, Carlini et al. (2023) divide images into non-overlapping $128 \times 128$ tiles and compute the maximum $\ell_2$ distance between corresponding tiles. We adopt the identical setting for both Stable Diffusion v1.4 and v2.0. As the detection performance of this metric achieves the best after full sampling steps, we only report the complete **50-step** results in Table 1.

- **(Ren et al., 2024)**: Based on the empirical observation that patterns in attention scores for specific tokens (termed as "trigger tokens") behaves differently in memorized samples, Ren et al. (2024) introduce the detection score $D$ and layer-specific entropy $E_{t=T}^l$ as primary indicators of memorization.

  The first metric $D$, which we refer to **Average Entropy (AE)** for intuitive notation, is defined as:

$$AE = \frac{1}{T_D} \sum_{t=0}^{T_D-1} E_t + \frac{1}{T_D} \sum_{t=0}^{T_D-1} |E_t^{\text{summary}} - E_T^{\text{summary}}|,$$

  where $E_t$ represents attention entropy, measuring the dispersion of attention scores across different tokens:

$$E_t = \sum_{i=1}^{N} -\overline{a}_i \log(\overline{a}_i).$$

  In addition, $E_t^{\text{summary}}$ is the entropy computed only on the summary tokens, and $T_D = \frac{T}{5}$ corresponds to the last $\frac{T}{5}$ steps of the reverse diffusion process used for memorization detection.

  The second metric, layer-specific entropy $E_{t=T}^l$, which we refer to **Layer Entropy (LE)**, is computed at the first

diffusion step and focuses on specific U-Net layers:

$$LE = \sum_{i=1}^{N} -\bar{a}_i^l \log(\bar{a}_i^l),$$

where $\bar{a}_i^l$ is the average attention score in layer $l$. For detection experiments, we follow the implementation and hyperparameter settings of Ren et al. (2024). The detection performance differences between our results in Table 1 and those reported in Ren et al. (2024) can be attributed to different choices of non-memorized prompts. Specifically, our evaluation uses prompts collected from diverse sources, whereas Ren et al. (2024) utilizes GPT-4 generated prompts that share similar characteristics. For comprehensive experimental details, we refer readers to Ren et al. (2024).

- **(Wen et al., 2024)**: Building on the insight that significant text guidance induces memorized samples during sampling, Wen et al. (2024) propose using the magnitude of predicted noise difference between conditional and unconditional noise. It is defined as:

$$\frac{1}{T} \sum_{t=1}^{T} \|\boldsymbol{\epsilon}_\theta(\mathbf{x}_t, c) - \boldsymbol{\epsilon}_\theta(\mathbf{x}_t, \emptyset)\|,$$

where $T$ denotes the number of timesteps, $c$ denotes the specific embedded prompt, and $\emptyset$ denotes empty string, equivalent to unconditional case. Recall that diffusion forward process $q_{t|0}(\mathbf{x}_t|\mathbf{x}_0) = \mathcal{N}(\sqrt{\alpha_t}\mathbf{x}_0, (1 - \alpha_t)\mathbf{I})$ and therefore,

$$\nabla_{\mathbf{x}_t} \log p_t(\mathbf{x}_t) = \mathbb{E}_{p_0(\mathbf{x}_0)} \left[ \nabla_{\mathbf{x}_t} \log q(\mathbf{x}_t|\mathbf{x}_0) \right] \approx \mathbb{E}_{p_0(\mathbf{x}_0)} \left[ -\frac{\boldsymbol{\epsilon}_\theta(\mathbf{x}_t)}{\sqrt{1 - \alpha_t}} \right] = -\frac{\boldsymbol{\epsilon}_\theta(\mathbf{x}_t)}{\sqrt{1 - \alpha_t}} = s_\theta(\mathbf{x}_t).$$

Thus,

$$\|s_\theta(\mathbf{x}_t, c) - s_\theta(\mathbf{x}_t)\| = \frac{1}{\sqrt{1 - \alpha_t}} \|\boldsymbol{\epsilon}_\theta(\mathbf{x}_t, c) - \boldsymbol{\epsilon}_\theta(\mathbf{x}_t, \emptyset)\|.$$

Consequently, Wen's metric can be defined as the norm of score differences as described in Section 4.3.

- **(Chen et al., 2024)**: Building on the observation that the end token exhibits abnormally high attention scores for memorized prompts, specifically highlighting the memorized region, Chen et al. (2024) leverage this attention score as a mask to amplify the detection of the **Partial Memorization (PM)** cases. We refer this metric as **Bright Ending (BE)** for short.

  In detail, Chen et al. (2024) multiply the attention mask $\mathbf{m}$ on Wen's metric:

$$BE = \frac{1}{T} \sum_{t=1}^{T} \|\left(\boldsymbol{\epsilon}_\theta(\mathbf{x}_t, c) - \boldsymbol{\epsilon}_\theta(\mathbf{x}_t, \emptyset)\right) \circ \mathbf{m}\| \bigg/ \left( \frac{1}{N} \sum_{i=1}^{N} m_i \right),$$

  where $N$ denotes for the number of elements in the mask $\mathbf{m}$, therefore the result is normalized by the mean of $\mathbf{m}$.
  We note that the attention mask $\mathbf{m}$ is obtainable at the final sampling step ($t = 1$). Therefore, to utilize **BE** as a detection metric, the model requires completion of all sampling steps. Consequently, in Table 1, we report experimental results using the complete **50-step** diffusion process.

  In addition, following the identical setup as Chen et al. (2024), we average attention scores from the first two downsampling layers of U-Net to obtain $\mathbf{m}$ for both Stable Diffusion v1.4 and v2.0. For additional details, refer to the original paper of Chen et al. (2024).

## D.2. Memorization Mitigation

**Details for Baseline Methods.** We provide details for each recent baseline mitigation algorithm. For every mitigation strategy, results are averaged over five generations per memorized prompt. Additionally, each baseline is evaluated using five different hyperparameter settings, which are described in detail below.

- **Random Token Addition (RTA) & Random Number Addition (RNA)**: Somepalli et al. (2023b) propose mitigation strategies that perturb prompts by adding arbitrary tokens or numbers. Following Wen et al. (2024), we insert tokens or numbers in quantities of {1, 2, 4, 6, 8} for both RTA and RNA.

- **(Ren et al., 2024)**: Ren et al. (2024) propose a mitigation strategy that involves masking memorization-inducing tokens and rescaling the attention scores of the beginning token using a hyperparameter $C$. After token masking, we evaluate the approach by varying $C$ within the range $\{1.1, 1.2, 1.25, 1.3, 1.5\}$ for both v1.4 and 2.0.

- **(Wen et al., 2024)**: As explained in Appendix D.1, Wen et al. (2024) propose a differentiable metric based on the norm of the difference between the conditional and unconditional scores. Since memorized prompts empirically exhibit a large magnitude for this term, Wen et al. (2024) optimize the text embedding by directly minimizing it.

  Wen et al. (2024) introduce $\ell_{target}$, a hyperparameter for early stopping, to prevent the text embedding from deviating significantly from its original semantic meaning. Following Wen et al. (2024), we investigate $\ell_{target}$ values ranging from 1 to 5 in Stable Diffusion v1.4. However, in v2.0, we found the generated results to be more sensitive. Therefore, for v2.0, we investigate $\ell_{target}$ values in $\{1, 1.25, 1.5, 1.75, 2\}$.

---

**Algorithm 2** SAIL pseudo-code

---

**Require:** Initialization $\mathbf{x}_T \sim \mathcal{N}(\mathbf{0}, \mathbf{I})$, Early stopping threshold $\ell_{thres}$, Score function $s(\cdot)$, Loss balancing term $\alpha$, Step size $\eta > 0$
**Ensure:** Set $\mathcal{L}_{\text{SAIL}} \leftarrow L_0$ {where $L_0 > \ell_{thres}$}
1: **while** $\mathcal{L}_{\text{SAIL}} > \ell_{thres}$ **do**
2:     Compute $s_\theta^\Delta(\mathbf{x}_T) \coloneqq s_\theta(\mathbf{x}_T, c) - s_\theta(\mathbf{x}_T)$;
3:     Normalize $s_\theta^\Delta(\mathbf{x}_T)$ with $\delta$ and compute $s_\theta^\Delta\big(\mathbf{x}_T + \delta s_\theta^\Delta(\mathbf{x}_T)\big)$;
4:     Compute SAIL objective:
5:     $\mathcal{L}_{\text{SAIL}}(\mathbf{x}_T) \coloneqq \big\| s_\theta^\Delta\big(\mathbf{x}_T + \delta \cdot s_\theta^\Delta(\mathbf{x}_T)\big) - s_\theta^\Delta(\mathbf{x}_T)\big\|^2 + \alpha\|\mathbf{x}_T\|^2$;
6:     Update initialization: $\mathbf{x}_T \leftarrow \mathbf{x}_T - \eta \nabla_{\mathbf{x}_T} \mathcal{L}_{\text{SAIL}}$;
7: **end while**

---

**Details for Our Method.** Algorithm 2 provides a pseudo-code for SAIL algorithm. While Algorithm 2 shows the case of optimizing a single $\mathbf{x}_T$, in practice, it can simultaneously search for several memorization-free candidates by collectively optimizing several initializations in a batch fashion.

To employ SAIL, we need to set $\alpha$ and $\ell_{thres}$. We set $\alpha = 0.05$ for Stable Diffusion v1.4 and $\alpha = 0.01$ for v2.0. In practice, we observe that the generated results are largely insensitive to $\alpha$, though keeping $\alpha$ sufficiently small helps balance the magnitude of two loss terms effectively. In addition, we investigate $\ell_{thres} \in \{7.6, 7.8, 8.2, 8.6, 9\}$ for v1.4 and $\{4, 4.5, 5, 5.5, 6\}$ for v2.0.

As the metric proposed by Wen et al. (2024) also captures sharpness, one may consider replacing the first term of $\mathcal{L}_{\text{SAIL}}(\mathbf{x}_T)$ with $\|s_\theta(\mathbf{x}_t, c) - s_\theta(\mathbf{x}_t)\|^2$. However, we empirically find that this alternative fails to converge and is therefore ineffective for mitigation. This may be due to the higher sensitivity of our proposed metric during the initial phase of generation.

**Details of prompts in Figure 6.** We provide full prompt details with a key prompt detail in **bold**, starting from top image.

- *The **Colbert** Report Gets End Date*

- ***Björk** Explains Decision To Pull Vulnicura From Spotify*

- ***Netflix** Hits 50 Million Subscribers*

- ***South Park**: The Stick of Truth Review (Multi-Platform)*

## E. Additional Qualitative Results for Memorization Mitigation

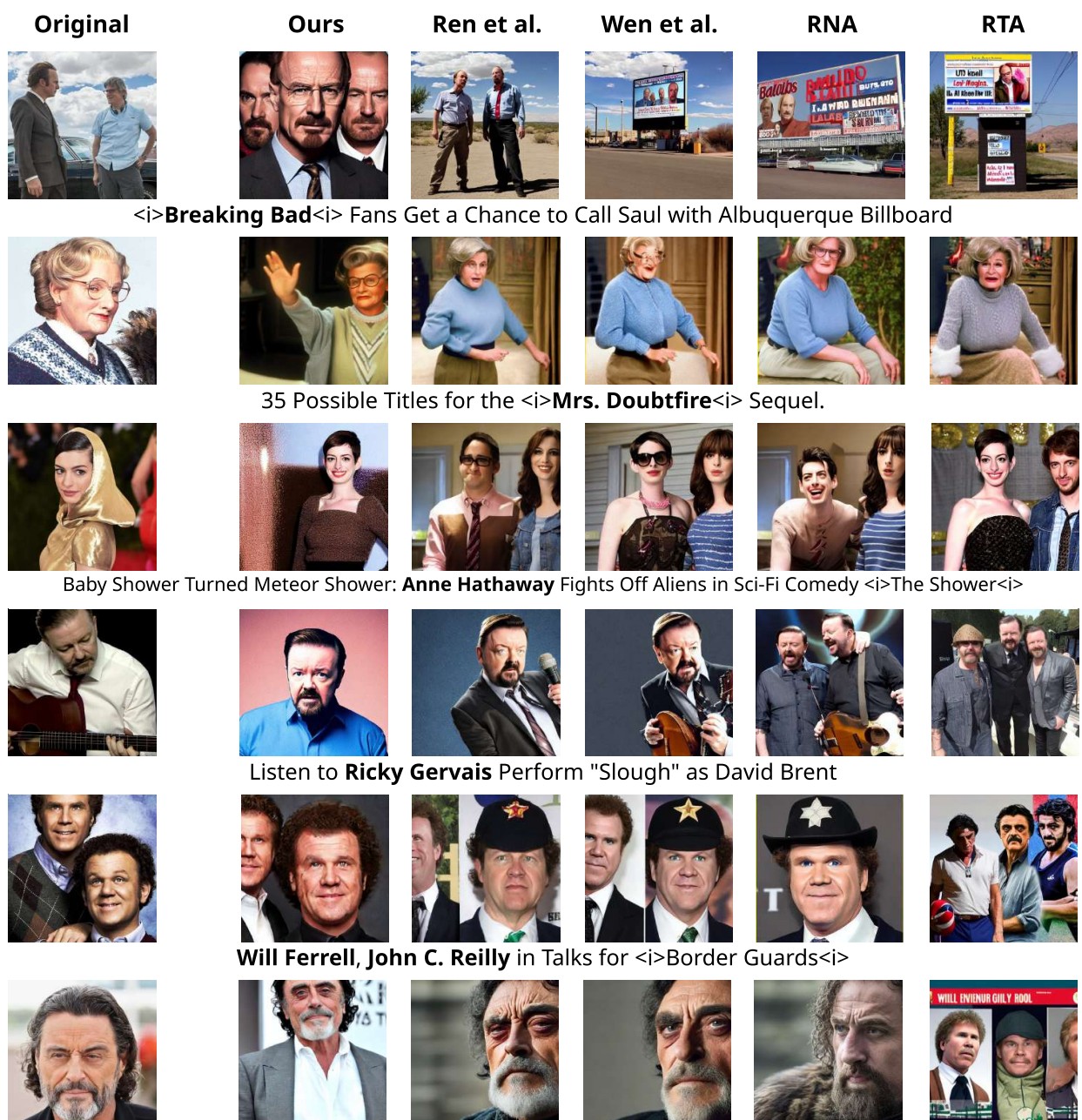

Figure 7: Additional qualitative results comparing SAIL with baseline methods. Original prompts are shown for each row with key elements in **bold**. All methods use identical initialization per prompt. SAIL effectively mitigates memorization while preserving prompt details, whereas baseline methods that modify text conditioning exhibit quality degradation.

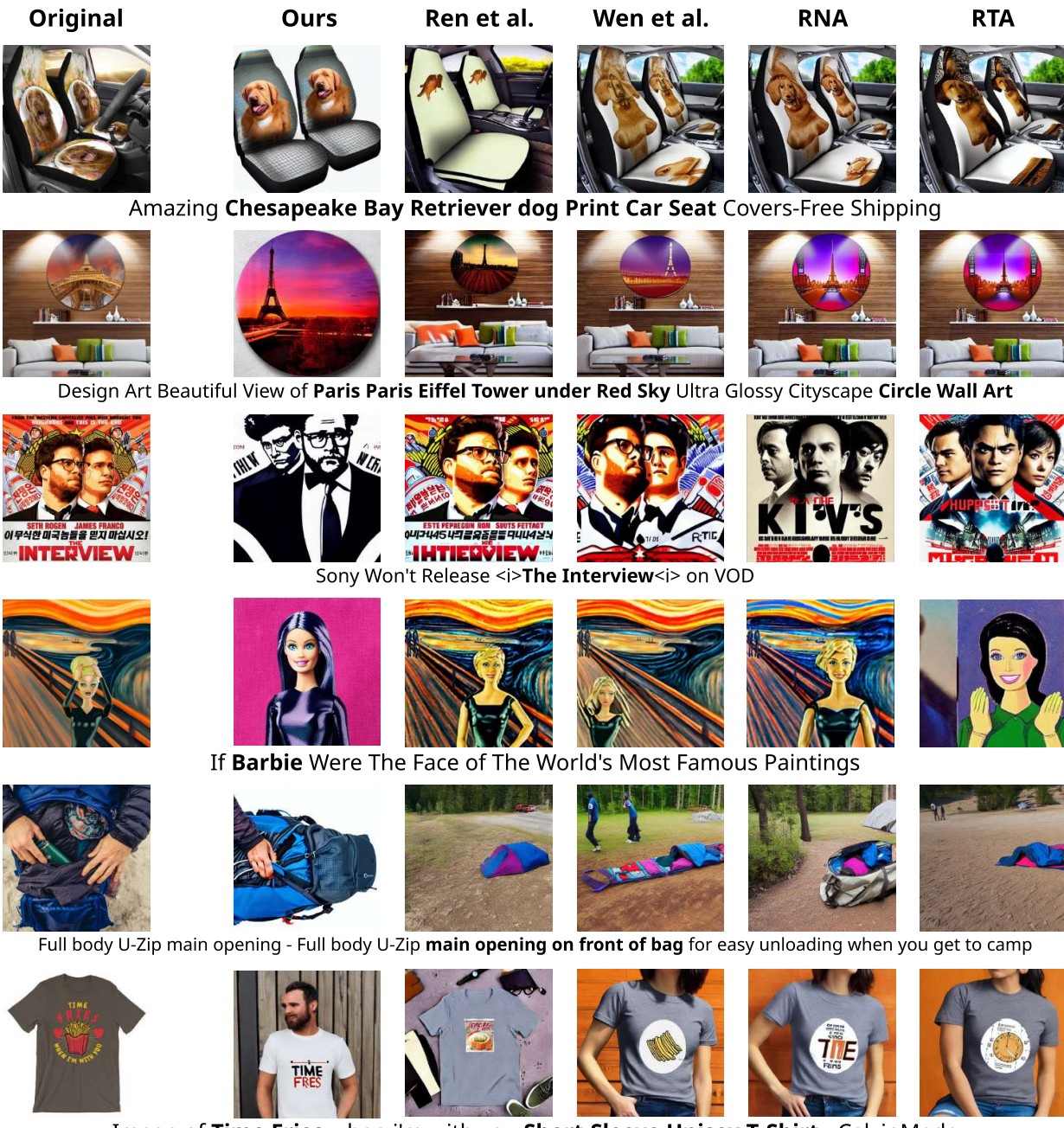

Figure 8: Additional qualitative comparison of SAIL against baseline methods. Each row shows original prompts with key elements in **bold**, and all methods share identical initialization per prompt. SAIL successfully mitigates memorization while preserving prompt details, whereas baseline methods with text conditioning modifications either degrade image quality or fail to mitigate memorization.

