# OpenReview forum: "Understanding and Mitigating Memorization in Generative Models via Sharpness of Probability Landscapes"
_ICML.cc/2025/Conference — ICML 2025 spotlightposter_

### Official Review · Reviewer_zVVR · 2025-03-07

**Overall Recommendation:** 3

**Summary:**

The paper presents a geometrical analysis of memorization in generative diffusion models based on the Hessian of their energy function around generated points. The idea follows naturally from recent results on the geometry of generative diffusion, which relate memorization and generalization to the spectrum of eigenvalues of the energy landscape. Based on these ideas, the paper introduces an effective and tractable method to detect memorized samples, which is shown to have high accuracy compared with established baseline. The authors also provide an initialization method that is shown to mitigate the generation of memorized examples.

**Claims And Evidence:**

The main claims are well supported both on an intuitive and on an experimental level. However, some of the theoretical considerations are somewhat handweavy and should be further elaborated. In particular, it would be important to clarify the relation between the trace of the Jacobian and the norm of the score function in the general non-Gaussian case.

**Essential References Not Discussed:**

[1] Achilli, Beatrice, et al. "Losing dimensions: Geometric memorization in generative diffusion." arXiv preprint arXiv:2410.08727 (2024).
[2] Ambrogioni, Luca. "In search of dispersed memories: Generative diffusion models are associative memory networks." Entropy 26.5 (2024): 381.
[3] Hoover, Benjamin, et al. "Memory in plain sight: A survey of the uncanny resemblances between diffusion models and associative memories." Associative Memory {\&} Hopfield Networks in 2023. 2023.
[4] Kadkhodaie, Zahra, et al. "Generalization in diffusion models arises from geometry-adaptive harmonic representations." arXiv preprint arXiv:2310.02557 (2023).

**Experimental Designs Or Analyses:**

The experiments and well executed and provide solid support to the claims.

**Methods And Evaluation Criteria:**

The experimental analysis is comprehensive and it provides robust results in favor of the main claims.

**Other Comments Or Suggestions:**

None

**Other Strengths And Weaknesses:**

The main strengths are in the ideas and in the experimental evaluations and results. The main weakness is in the lack of rigor in several theoretical points. However, I am of the opinion that the main ideas can be fully formalized in a very elegant way and I would encourage the authors to work on obtaining more general theoretical results.

**Questions For Authors:**

None

**Relation To Broader Scientific Literature:**

I do think that the main innovation of the paper is the introduction of the Hessian upscaling method. In general, I appreciated how this work used competently and effectively several ideas that were floating in the geometric analysis of diffusion models through their Jacobian spectra. A very related piece of literature is given in [1], where the onset of (geometric) memorization is identified by the closure of spectral gaps in the Jacobian, which can be connected directly to the onset of sharpness in the trace.
It would also be useful to discuss the connections with modern Hopfield networks, which has been shown to be equivalent to diffusion models in the memorization regime [2,3]. Finally, the analysis should be connected with the similar results in [4], which studies the onset of generalization through a similar analysis of the Jacobi matrix.
I do think that the main innovation of the paper is the introduction of the Hessian upscaling method. In general, I appreciated how this work used competently and effectively several ideas that were floating in the geometric analysis of diffusion models through their Jacobian spectra. A very related piece of literature is given in [1], where the onset of (geometric) memorization is identified by the closure of spectral gaps in the Jacobian, which can be connected directly to the onset of sharpness in the trace.

It would also be useful to discuss the connections with modern Hopfield networks, which has been shown to be equivalent to diffusion models in the memorization regime [2,3]. These works already highlighted some of the ideas discussed here, for example the fact that memorized states should have sharp energy wells around them.

Finally, the analysis should be connected with the similar results in [4], which studies the onset of generalization through a similar analysis of the Jacobi matrix.

**Theoretical Claims:**

The intuitive ideas are very well motivated but the paper would have benefited from a more rigorous theoretical approach in the non-Gaussian case. I think it should be possible to obtain general formulas that relate the norm of the score to the trace in the general case, possibly using a second order Tweedie's Formula.
It would be nice to see if the authors can derive a general formula. I think it should also be possible to provide more rigorous theoretical motivations for the upscaling formula.

---

> ### Author Rebuttal · Authors · 2025-03-31
>
> > Relation between the trace of the Jacobian and the norm of the score function in the non-Gaussian case
>
> Thank you for the insightful comment regarding the relation between the trace of the Jacobian and the score norm in the non-Gaussian case.
>
> We found that Lemma 4.1 is indeed generalizable beyond the Gaussian case under mild boundary conditions [1] (e.g., $\lim_{|\mathbf{x}| \to \infty} p(\mathbf{x}) = 0$ and $\lim_{|\mathbf{x}| \to \infty} p(\mathbf{x}) \nabla \log p(\mathbf{x}) = 0$), under which the identity
>
> $\mathbb{E}[\||s(\mathbf{x}) \||^2] = -\mathbb{E}[\mathrm{tr}(H(\mathbf{x}))]$
>
> holds in general where the trace is now in expectation.
>
> We also appreciate the suggestion to consider Tweedie's formula for Lemma 4.3. We are currently revisiting the proof and will consider incorporating it into the formulation. Regarding Lemma 4.2, which quantifies the gap between conditional and unconditional distributions, we chose Gaussian assumption to ensure tractability and interpretability. This choice is supported by prior work [2] and the empirical trends shown in Figure 4, making our approximation both practical and well-justified.
>
> - [1] Hyvärinen, "Estimation of Non-Normalized Statistical Models by Score Matching", JMLR, 2005.
> - [2] Wang et al. "The unreasonable effectiveness of gaussian score approximation for diffusion models and its applications." TMLR, 2024.
>
> > Connections with modern Hopfield networks
>
> We sincerely thank the reviewer for highlighting important theoretical connections between our paper and recent work on modern Hopfield networks. As pointed out by the reviewer, [1] demonstrates that large class of diffusion models asymptotically yield energy landscapes equivalent to modern Hopfield networks, where memorized states correspond to local minima. [2] similarly presents an insightful perspective by interpreting diffusion models as a extension of associative memory retrieval, showing that the iterative denoising process parallels recurrent energy minimization in Hopfield-like networks. Their framework highlights how the dynamic updating of latent variables in diffusion models can be cast as an attractor-based process, reinforcing the broader view of diffusion as a form of associative memory.
>
> Our paper shares the central theoretical idea-that memorized states are sharp minima-but differs in scope and methodology. While these referenced studies focus on establishing theoretical equivalences and analyzing associative memory capacity, our contribution lies in explicitly quantifying sharpness via Hessian eigenvalues and score norms as an early-stage utility tool to detect and mitigate memorization during model inference. We greatly appreciate the reviewer's insightful suggestion to clarify these theoretical connections, as this strengthens the context and clarity of our work.
> - [1] Ambrogioni "In search of dispersed memories: …", Entropy, 2024
> - [2] Hoover et al. "Memory in Plain Sight:...", NeurIPS Workshop, 2023.
>
>
> > Connection with the eigenbasis framework
>
> We sincerely appreciate your valuable suggestions, particularly regarding closely related studies that enrich the theoretical context of our work. The concurrent work [1] rigorously characterizes geometric memorization through analysis of the score function's Jacobian eigenstructure. The authors identify memorization onset when spectral gaps between singular values close, indicating the manifold's tangent-space structure has collapsed. This eigenbasis approach aligns with our central idea-that memorization corresponds to increased local curvature, which implies very small local variance in the density. We find their elegant mathematical formulation very insightful and acknowledge the deep theoretical contributions they've made to understanding generative model memorization.
>
> However, while their detailed eigenanalysis can be computationally prohibitive for large-scale diffusion models like Stable Diffusion, our approach simplifies the process by using the Hessian score product as a summary statistic. This streamlined method maintains theoretical coherence while efficiently monitoring and mitigating memorization in high-dimensional generative models.
>
> Similarly, [2] demonstrates how strong generalization naturally emerges through geometry-adaptive harmonic representations. Their work shows that optimal denoisers implicitly operate within a geometry-adaptive basis, explaining how diffusion models achieve generalization without exponentially large datasets. While they characterize the adaptive basis underlying generalization, our contribution focuses specifically on sharpness as a direct, practical measure of memorization, complementing their basis-oriented theoretical analysis. We are grateful for these insightful references to the reviewer and will update our manuscript accordingly.
> - [1] Achilli, Beatrice, et al. "Losing dimensions: ….", arxiv, 2024
> - [2] Kadkhodaie et al. "Generalization in diffusion models arises …", ICLR, 2024.

---

### Official Review · Reviewer_SF4v · 2025-03-09

**Overall Recommendation:** 4

**Summary:**

This paper proposes to understand and mitigate the memorization of diffusion models from the perspective of the sharpness of probability landscapes. More specifically, it first shows that the large negative eigenvalues of the Hessian matrix, which reflects the sharpness, can indicate the risk of memorization. It then proposes a computationally efficient metric (i.e., Hessian trace and score norm) to measure the sharpness. The authors also show that the popular Wen’s metric can be explained from the aspect of sharpness and enhances Wen’s metric to enable the early-stage detection of memorization. Finally, the authors develop a sharpness-aware initialization method to mitigate the memorization. Experimental results on MNIST and Stable Diffusion reveal that the memorization of the diffusion model can be detected and mitigated by the sharpness-based method provided by the authors.

## Update After Rebuttal
I think my concerns are addressed by the rebuttal. Therefore, I prefer to maintain my original rating of 4, showing that I tend to accept this paper.

**Claims And Evidence:**

I believe that the claims made in the submission are supported by clear and convincing evidence.

**Essential References Not Discussed:**

I think the essential references have been covered by the authors.

**Experimental Designs Or Analyses:**

I have checked the experimental designs and analyses. I think the evaluation is valid.

**Methods And Evaluation Criteria:**

The proposed method and evaluation are reasonable for detecting and mitigating the memorization in diffusion models.

**Other Comments Or Suggestions:**

N/A

**Other Strengths And Weaknesses:**

The strengths of the paper are listed as follows.
1. The paper is well-written and well-developed, making it easy to follow. Even if the readers have little background knowledge, they can easily get the key points of the paper.
2. All of the important claims in the paper are supported by both theoretical analysis or proof and empirical evaluation.
3. The idea is novel. To the best of my knowledge, it is the first work to understand and mitigate the memorization of the diffusion model from the perspective of sharpness.
4. The proposed method is practical. The proposed methods to detect and mitigate memorization are computationally affordable and can be used in real applications.

The other weaknesses of the paper are listed as follows.
1. It would be better if the authors could evaluate their methods on more datasets. However, given the theoretical analysis from the authors, I think it is just a minor weakness.
2. For section 4.4, the existing empirical results in Table 1 do not show an obvious improvement over Wen’s metric at step 1. It would be better if the authors could find a case where Wen’s metric performs badly to support the importance of upscaling.

**Questions For Authors:**

1. In Section 5.1, how do you solve the formulated objective problem for sharpness-aware initialization?
2. In the right part of Figure 3, did you miss the negative sign for the lower part values in the y-axis?
3. In the left part of Figure 6, for the proposed method, why can the CLIP score not reach larger values (e.g., > 0.26) like the other baselines?

**Relation To Broader Scientific Literature:**

I think this paper is highly relevant to Wen’s metric [1] and the LID work [2, 3].

[1]. Wen, Y., Liu, Y., Chen, C., and Lyu, L. Detecting, explaining and mitigating memorization in diffusion models. In ICLR, 2024.
[2]. Ross, B. L., Kamkari, H., Wu, T., Hosseinzadeh, R., Liu, Z., Stein, G., Cresswell, J. C., and Loaiza-Ganem, G. A geometric framework for understanding memorization in generative models. arXiv preprint arXiv:2411.00113, 2024.
[3]. Kamkari, H., Ross, B. L., Hosseinzadeh, R., Cresswell, J. C., and Loaiza-Ganem, G. A geometric view of data complexity: Efficient local intrinsic dimension estimation with diffusion models. In ICML 2024 Workshop on Structured Probabilistic Inference & Generative Modeling, 2024.

**Theoretical Claims:**

To the best of my knowledge, I think the proofs for the theoretical claims (Lemma 4.1-4.3) should be correct.

---

> ### Author Rebuttal · Authors · 2025-03-31
>
> > Evaluate methods on more datasets.
>
> Thank you for the suggestion. For our Stable Diffusion experiments, we adopted the established benchmark of known memorized prompts introduced by [1], which has become a standard dataset in the current literature [2-4]. In line with prior work, we made an effort to comprehensively include all verbatim categories within this benchmark.
> - [1] Webster, "A reproducible extraction …" arXiv. 2023.
> - [2] Wen et al. "Detecting, explaining, and mitigating …", ICLR. 2024.
> - [3] Ren et al. "Unveiling and mitigating memorization …", ECCV, 2024.
> - [4] Chen et al. "Exploring local memorization …", ICLR, 2025.
>
> > Empirical results in Table 1 (Comparison with Wen’s metric at step 1)
>
> Thank you for the suggestion. We would like to highlight the advantages of our upscaling method from two perspectives: detection and mitigation.
>
> **Computational Cost**
>
> Our metric offers clear computational benefits in detection. As shown in Table 1, our approach consistently achieves comparable or superior performance while requiring substantially fewer sampling steps (“Steps”) and fewer simultaneous generations (“n”) compared to Wen’s metric. This efficiency is especially valuable in practical scenarios such as real-time detection.
>
> Below, we present a comparison of computation time (in seconds) between Wen’s metric and ours using Stable Diffusion v1.4. We highlight in **bold** the entries where both methods achieve equivalent AUC performance.
>
> |-|Step 1|Step 5|Step 50|
> |-|-|-|-|
> |Wen et al. (n=1)|0.233|0.431|3.211|
> |Wen et al. (n=4)|0.728|1.323|11.25|
> |Wen et al. (n=16)|1.326|**1.955**|**16.55**|
> |||||
> |Ours (n=1)|0.412|–|–|
> |Ours (n=4)|**0.882**|–|–|
>
> As shown above, our method achieves similar or better performance at a clearly lower computational cost. We also expect further efficiency gains if the JVP operation is optimally integrated into libraries such as Hugging Face.
>
> **Harder Detection Cases in SD v2.0:**
>
> Unlike Stable Diffusion v1.4, which includes both Exact and Partial memorized samples, v2.0 contains only Partial memorized prompts, making the detection task more challenging. The more pronounced performance gap between our method and Wen’s metric in this setting empirically demonstrates the effectiveness of our upscaling approach.
>
> **Mitigation - SAIL Optimization:**
>
> Our metric is also crucial for SAIL. Although Wen's metric works well for detection, using it in the SAIL objective (line 360) resulted in optimization failure. Our approach, however, successfully achieved stable convergence and effective mitigation by amplifying differences in the early stages.
>
> > How to solve SAIL in Section 5.1
>
> Thank you for the question. While we provide detailed pseudocode for SAIL in Appendix D.2, we are happy to explain it here as well.
>
> The SAIL objective involves two forward passes at the initial sampling step ($t=T-1$).
> - In the first pass, we compute the score difference $s_{\theta}^\Delta(\mathbf{x}_T) $.
> - We then perturb the initialization slightly in the direction of this score difference, i.e.,
> $\mathbf{x}_T +  \delta \cdot s\_\theta^\Delta (\mathbf{x}_T) $,
> - and perform a second forward pass to obtain
> $s_{\theta}^\Delta\bigl(\mathbf{x}_T + \delta \cdot s\_{\theta}^\Delta(\mathbf{x}_T)\bigr) $.
>
> With these components, we optimize the initial noise $ \mathbf{x}_T$ sampled from an isotropic Gaussian using the SAIL objective (refer to line 373).
>
> This optimization is lightweight and typically converges within two to three iterations on average, depending on the threshold hyperparameter  $ \ell_{\text{thres}} $. SAIL also supports batch-wise execution, allowing multiple initializations to be optimized in parallel. Please feel free to let us know if any part requires further clarification.
>
> > In Figure 6 (left), for SAIL, why can the CLIP score not reach larger values like the other baselines?
>
> We appreciate your valuable question. It is indeed possible to achieve higher average CLIP scores with SAIL by adjusting its threshold hyperparameter $\ell_{\text{thres}}$.
>
> However, we would like to clarify that, empirically, algorithms achieving high CLIP scores alongside high SSCD scores (e.g., > 0.35) often produce outputs that are only superficially altered. These include blurry or partially contaminated memorized images, which are difficult to consider genuinely mitigated.
>
> For all baseline methods, we carefully selected hyperparameters based on their original papers to ensure a fair comparison. In contrast, for SAIL, we prioritized configurations that provide strong memorization mitigation. Notably, across extensive hyperparameter sweeps, SAIL consistently achieved the lowest points on the SSCD -CLIP performance trade-off curves in both Stable Diffusion v1.4 and v2.0.
>
> >  In the Figure 3 (right), did you miss the negative sign in the y-axis?
>
> Thank you for pointing this out. The reviewer is correct and we will revise it accordingly in the final version.

---

> > ### Comment · Reviewer_SF4v · 2025-04-05
> >
> > Thanks for the detailed response from the authors. I think my concerns are addressed by the rebuttal. Therefore, I prefer to maintain my original rating of 4, showing that I tend to accept this paper.

---

> > > ### Author Response · Authors · 2025-04-06
> > >
> > > We sincerely thank the reviewer for positively recognizing the contributions of our work. We greatly appreciate your valuable time, effort, and insightful feedback, which will help us further improve our manuscript.
> > >
> > > Sincerely,\
> > > The authors

---

### Official Review · Reviewer_qYB7 · 2025-03-10

**Overall Recommendation:** 3

**Summary:**

To alleviate the memory effect of the diffusion model, this paper proposes a sharpness-based detection metric and develops an effective mitigation strategy based on this metric. The strengths of this paper lie in its clarity and the progressive experiments and theoretical analysis that illustrate the rationale and effectiveness of the proposed method. The proposed mitigation strategy outperforms existing methods, while requiring no additional modifications on texts and model architecture.

**Claims And Evidence:**

1. The metric analyzed in this paper is almost identical to that in [1], with the main difference being whether it analyzes the first-order properties of the score function or the denoiser. The difference between the two lies only in a constant, resulting in nearly no distinction in properties such as the Jacobian. Additionally, conclusions like "smoother regions tend to yield non-memorized images" (line 375) also appear in [1]. However, this paper does not discuss that work, which weakens its contribution.

2. The key motivation of this paper (see Lines 160-164) lacks sufficient evidence. Although the expected phenomenon is observed on toy data such as MNIST, merely observing the phenomenon does not justify the validity of this key motivation. Could additional theoretical explanations or further analysis be provided?

[1] Wang, Hanyu, Yujin Han, and Difan Zou. "On the discrepancy and connection between memorization and generation in diffusion models." ICML 2024 Workshop on Foundation Models in the Wild. 2024.

**Essential References Not Discussed:**

See Claims And Evidence.

**Experimental Designs Or Analyses:**

1. The right-side image in Figure 2 shows that the eigenvalues of memorized and non-memorized samples at the initial sampling step are not very large. A similar issue appears in Figure 3, where at \( t = T-1 \), the differences between different memorization categories are not significant enough. This undermines the necessity of monitoring memorization throughout the entire sampling process, making the advantage over works like LID less compelling.

2. Table 1 shows that the proposed method performs very similarly to Wen’s metric. For example, at \( T = 1 \), the AUC difference is only between \( 1e{-3} \) and \( 1e{-2} \). Moreover, the proposed metric requires the additional computation of the Hessian matrix.

3. The proposed mitigation strategy may not be applicable to stochastic sampling methods such as SDE.

**Methods And Evaluation Criteria:**

1. Works like [1,2] have already used curvature properties to discuss memorization, and this paper seems to upgrade Wen’s metric while introducing the additional assumption that \( x_t \) follows a Gaussian distribution.

2. To facilitate the computation of the proposed metric, the authors introduce several approximations. Such as Lemma 4.1, which connects the trace with the norm of the score function, Lemma 4.2, which links Wen’s Metric to the proposed Sharpness Measure, or Lemma 4.3, all assume a Gaussian distribution. These assumptions raise concerns regarding the discrete diffusion model's reverse process and prompt a crucial question: how accurate is the improved metric in real-world scenarios?

[2] Kamkari, H., Ross, B. L., Hosseinzadeh, R., Cresswell, J. C., and Loaiza-Ganem, G. A geometric view of data complexity: Efficient local intrinsic dimension estimation with diffusion models. In ICML 2024 Workshop on Structured Probabilistic Inference & Generative Modeling, 2024.

**Other Comments Or Suggestions:**

N/A

**Other Strengths And Weaknesses:**

N/A

**Questions For Authors:**

N/A

**Relation To Broader Scientific Literature:**

This paper proposes a new metric and method to detect the memorization of diffusion models. The underlying principle, such as the use of sharpness, has appeared in previous work. The authors leverage this property and further improve the existing Wen’s metric, achieving enhancement.

**Theoretical Claims:**

N/A

---

> ### Author Rebuttal · Authors · 2025-03-31
>
> > Connection to [1].
>
> We thank the reviewer for highlighting the connection to [1], which we will properly acknowledge. While both works study memorization through geometric properties of the density, there are key differences:
> - [1] focuses on first-order smoothness via comparisons between trained and oracle scores, whereas our analysis emphasizes second-order geometry through the trained Hessian and explicitly measures sharpness by observing distribution of eigenvalues.
> - We also demonstrate how specific initial latents lead to memorization by consistently mapping into high-curvature regions, further amplified by text conditioning.
>
> We believe these distinctions clearly position our contributions as complementary to [1], enriching the understanding of memorization in generative models.
>
> > Evidence of key motivation
>
> We appreciate the concern about the generalizability of our observed phenomenon. While we use MNIST for clarity and intuition, our experimental design extends well beyond this toy dataset. As shown in Figure 3, the key phenomenon, sharpness correlating with memorization, is present in modern, large-scale models like Stable Diffusion.
>
> This progression from simple to complex domains actually strengthens our motivation by showing that the behavior persists across different datasets. Figure 5 further supports this finding, revealing consistent patterns in the Hessian spectra between memorized and non-memorized samples.
>
> > Works like [1,2] already used curvature properties to discuss memorization.
>
> We acknowledge that prior works [1, 2] have explored memorization and data complexity through curvature-related concepts. However, our contribution advances beyond these studies in both scope and methodology. While [2] examines curvature only at the final generation stage, we introduce a dynamic framework that analyzes sharpness throughout the entire diffusion process across all timesteps. This continuous perspective enables earlier detection and mitigation of memorization, which we believe are novel and practically valuable contributions.
>
> > Gaussian Assumptions
>
> Empirical evidence from recent work [3] shows that diffusion models exhibit approximately Gaussian score behavior in early and intermediate steps, supporting the validity of our Gaussian-based assumptions.
>
> As shown in Figure 4, key metrics like the negative Hessian trace align well with the score norm and Hessian-score product, confirming that our theoretical approximations hold in practice.
>
> > Small differences in eigenvalues
>
> We understand your concern. While eigenvalue differences at the initial step are small, they become effective when aggregated (e.g., sum or cubed sum). As shown in Table 1, both our metric and Wen’s metric, which measure these statistics, perform well at Step 1. This confirms the usefulness of early-stage eigenvalue signals.
>
> Regarding LID, we excluded it from our detection experiments since it requires full sampling steps, making direct comparison unfair. For reference, we provide our LID results at Step 50:
> - SD v1.4: AUC = 0.974 (n=1), 0.992 (n=4); TPR@1%FPR = 0.184 (n=1), 0.824 (n=4)
> - SD v2.0: AUC = 0.972 (n=1 and n=4); TPR@1%FPR = 0.470 (n=1), 0.216 (n=4)
>
> Compared to our method in Table 1, LID yields significantly lower TPR@1%FPR, especially in SD v2.0. This highlights the effectiveness of our sharpness-based approach, which is essential to the success of our mitigation strategy (Figure 6).
>
> > Performance \& Time cost compared to Wen's metric
>
> Thank you for pointing this out. While our method involves computing a Hessian-score product, the additional cost is minimal due to efficient JVP implementations in standard libraries.
>
> Our metric provides clear computational benefits. As Table 1 demonstrates, it achieves equal or superior AUC compared to Wen's metric with fewer sampling steps ("steps") and generations ("n"). Using SD v1.4, we compare the runtime (in seconds) between these metrics below, with **bold** entries indicating where both methods achieve equivalent AUC.
>
> |-|Step 1|Step 5|Step 50|
> |-|-|-|-|
> |Wen et al. (n=1)|0.233|0.431|3.211|
> |Wen et al. (n=4)|0.728|1.323|11.25|
> |Wen et al. (n=16)|1.326|**1.955**|**16.55**|
> |||||
> |Ours (n=1)|0.412|–|–|
> |Ours (n=4)|**0.882**|–|–|
>
> Our metric is also critical for SAIL. While Wen’s metric is effective for detection, we found that using it as SAIL objective (line 360) led to optimization failure. In our case, the amplified early-stage differences enabled stable convergence and effective mitigation.
>
> > SAIL may not be applicable to SDE.
>
> While most of prior works focus on ODE samplers for analysis, we believe SAIL can still be applied to SDE samplers in expectation by selecting good initializations. We agree this is a valuable direction for future work.
>
> [1] Wang et al. "On the discrepancy and connection...", ICML Workshop, 2024\
> [2] Kamkari et al. "A geometric view of data...", ICML Workshop, 2024\
> [3] Wang et al. "The Unreasonable Effectiveness...", TMLR, 2024.

---

> > ### Comment · Reviewer_qYB7 · 2025-04-02
> >
> > Thank you for the response. Most of the authors' replies addressed my concerns, so I have raised my score to 3.
> >
> > Why not a higher score? Regarding the connection to [1], I partially agree with the authors. Specifically, [1] also analyzes first-order properties of both the oracle model $\epsilon^*$ and the trained model $\epsilon_\theta$, including analysis about
> > eigenvalues (Fig. 3), while this paper analyzes second-order properties of $\log p$ (lines 150–155). These two analyses are essentially equivalent, given the fact that the difference between $\epsilon_{\theta}$ and $\nabla \log p$ is only a scaling factor of $-\frac{1}{\sigma_t}$. I agree with the authors that this work includes additional experimental findings (e.g., discussions on initial latents), but the same proposed metric reduces the contribution of this work.

---

> > > ### Author Response · Authors · 2025-04-04
> > >
> > > Thank you for your thoughtful and constructive comments, and for recognizing the contributions of our work. We appreciate your point regarding the connection to [1] and will ensure it is properly acknowledged and clarified in the final version. Thank you again for your valuable feedback and for helping improve the quality of our submission.

---

### Official Review · Reviewer_TT7u · 2025-03-19

**Overall Recommendation:** 4

**Summary:**

This paper studies the memorization phenomenon in diffusion models, which is a crucial task that is well-motivated by its practical significance in privacy preservation in the era of GenAI. This paper discovers a new pattern that can differentiate memorized and non-memorized generations of diffusion models that is based on the sharpness of the log probability density, quantified by the Hessian of the log probability, serving as a new detection strategy for the memorized generations. Also, it shows the relevance of this pattern with the existing pattern found by Wen et al. Armed with such a finding, this paper proposes a mitigation strategy named SAIL that can efficiently mitigate memorization while being more effective (better text alignment under the same privacy level) than existing baselines. Experiments are conducted on a 2D toy dataset, MNIST, and Stable Diffusion’s LAION dataset.

## update after rebuttal

I have no further questions, so I keep my original rating of accept.

**Claims And Evidence:**

This paper’s claims are well-supported by its analysis, either as visualizations or in the form of math proofs, and its superior experimental results.

**Essential References Not Discussed:**

The paper has provided a comprehensive discussion of the core related papers.

**Experimental Designs Or Analyses:**

The design of evaluating the proposed detection and mitigation strategies is sound, which follows the baseline that is shown in Table 1 and Figure 6.

**Methods And Evaluation Criteria:**

The benchmark datasets follow the existing baselines, which makes their comparisons well-justified for the task that they address.

**Other Comments Or Suggestions:**

N/A

**Other Strengths And Weaknesses:**

Please see the previous sections.

**Questions For Authors:**

Overall, this is a solid paper to me and I would like to recommend acceptance.

**Relation To Broader Scientific Literature:**

This paper specifically contributes to the field of understanding and addressing memorization issues in diffusion models. I believe this paper contributes to both theoretical and empirical aspects.

**Theoretical Claims:**

I have carefully checked the equations and claims in the main paper and observed no issues. Nevertheless, I am uncertain about the detailed proofs in the Supplementary Material.

---

> ### Author Rebuttal · Authors · 2025-03-31
>
> Thank you for your positive review and for recognizing our contributions to this topic. We sincerely appreciate the time and effort you dedicated to reviewing our work. Please do not hesitate to reach out with any further questions or suggestions.

---

### Decision · Program_Chairs · 2025-05-01

**Decision:**

Accept (spotlight poster)

**Comment:**

This paper studies the important and interesting problem of memorization in diffusion models using ideas and tools from geometry to analyse the sharpness of the probability landscape. The authors’ ideas build on recent literature that has also taken a geometric approach to understanding diffusion models, and therefore is well-placed in contemporary work. The authors propose a metric that can measure sharpness and relate this to an existing metric to enhance how it can indicate whether an image is likely to be memorized. Methods to mitigate memorization are also given.

The reviewers raised several points about the paper, including some additional papers that could be discussed, request to expand on the experimental results, and demonstrate more clearly where their work improves over Wen’s metrics. However, these points were not major obstacles for the submitted work, and several reviewers replied that they were satisfied by the discussions. Hence, it is clear that this paper should be recommended for acceptance.